# DIFFUSION MODELS AS DATASET DISTILLATION PRIORS

**Duo Su**[1], **Huyu Wu**[2], **Huanran Chen**[1], **Yiming Shi**[3], **Yuzhu Wang**[4],

**Xi Ye**[1], **Jun Zhu**[1] *

[1]Dept. of Comp. Sci. & Tech., BNRist Center, THU-Bosch ML Center, Tsinghua University.
[2]Institute of Computing Technology, CAS.
[3]University of Electronic Science and Technology of China.
[4]South China University of Technology.
https://suduo94.github.io/Diffusion-As-Priors

## ABSTRACT

Dataset distillation aims to synthesize compact yet informative datasets from large ones. A significant challenge in this field is achieving a trifecta of diversity, generalization, and representativeness in a single distilled dataset. Although recent generative dataset distillation methods adopt powerful diffusion models as their foundation models, the inherent representativeness prior in diffusion models is overlooked. Consequently, these approaches often necessitate the integration of external constraints to enhance data quality. To address this, we propose Diffusion As Priors (DAP), which formalizes representativeness by quantifying the similarity between synthetic and real data in feature space using a Mercer kernel. We then introduce this prior as guidance to steer the reverse diffusion process, enhancing the representativeness of distilled samples without any retraining. Extensive experiments on large-scale datasets, such as ImageNet-1K and its subsets, demonstrate that DAP outperforms state-of-the-art methods in generating high-fidelity datasets while achieving superior cross-architecture generalization. Our work not only establishes a theoretical connection between diffusion priors and the objectives of dataset distillation but also provides a practical, training-free framework for improving the quality of the distilled dataset.

## 1 INTRODUCTION

Data undeniably functions as the "primordial fuel" that drives modern AI systems. This critical resource provides large models with foundational knowledge, spatiotemporal comprehension, visual awareness, and pattern recognition capabilities (Brown et al., 2020; Qin et al., 2025). Despite this, data faces depletion as exponentially scaling models rapidly consume finite human-generated data, persisting as a bottleneck in advancing next-generation large models (Muennighoff et al., 2023; Villalobos et al., 2024). Current industry practices suffer dual burdens: insufficient data and expensive human annotation costs. Fortunately, synthetic data emerges as a renewable alternative capable of powering AI development at scale (Jordon et al., 2022; Liu et al., 2024). While large models can generate samples in arbitrary categories and sizes, unfiltered synthetic data poses two critical risks: *1) Data Quality Limitations* encompassing distribution drift and semantic mismatch (Alaa et al., 2022; Yang et al., 2024). *2) Training Hazards*, where flawed data patterns propagate through error amplification, triggering failures like model collapse (Shumailov et al., 2024; Dohmatob et al., 2024). Therefore, generating high-quality synthetic data remains a challenging task.

Recent advances in dataset distillation (DD) offer a promising solution to the above challenges by generating highly compact datasets while preserving critical features often obscured in real-world data (Wang et al., 2018). In parallel, diffusion models (DMs) have emerged as state-of-the-art generative methods due to their ability to accurately model the entire dataset distribution through score function estimation (Song et al., 2021). As a result, DMs have been adopted as foundation models

---

*Corresponding Author

for DD, giving rise to generative DD (Gu et al., 2024; Su et al., 2024). Leveraging priors acquired from well-trained DMs, distilled samples maintain diversity and fidelity, achieving competitive accuracy with up to $10\times \sim 200\times$ reduction in training size (Chen et al., 2025). Although encouraging, a theoretical analysis remains underdeveloped, which raises the following questions about the diffusion priors in generative DD methods.

**Do the priors in vanilla DMs satisfy the requirements for DD?** To answer this, we align the desired properties of distilled datasets with the priors captured by DMs via the original score function. From the perspective of log-likelihood estimation and evaluation metrics (e.g., FID, IS), we observe that the inherent diversity and generalization priors in vanilla DMs can yield higher-quality synthetic data. Naturally, the main challenge shifts to enhancing the representativeness of synthetic data, which is still not embodied in vanilla sampling pipelines. Previous approaches attempt to address this by imposing external representativeness constraints (Santiago et al., 2025; Chen et al., 2025). However, we argue that such constraints are unnecessary and introduce additional complexity. Thus, we raise the next question.

**Are there unused priors in DMs that could benefit DD?** Inspired by the diffusion classifiers (Chen et al., 2024a;b), we posit that the feature extraction capability inherent in a well-trained diffusion model itself constitutes a representativeness prior highly relevant to DD. We hypothesize that high representativeness corresponds to high similarity between synthetic and original data in the representation space. To formalize this, we employ the Mercer kernel, a specific type of kernel function (Zaanen, 1964), to quantify the similarity within feature spaces. The Mercer kernel provides us with mathematical guarantees of convexity and tractability in optimization, ensuring that the representativeness prior is computationally feasible. Empirically, we define the representativeness score function as an energy function based on Mercer kernel, which allows us to inject the unused representativeness prior into the distilled data through guided sampling.

We propose **D**iffusion **A**s **P**riors (**DAP**) and apply it to datasets of varying scales, including large-scale ImageNet-1K (Deng et al., 2009) and its small subsets. Both quantitative and qualitative results show that DAP significantly enhances the quality of distilled datasets. It validates the theoretical connections between diffusion priors and DD task, while achieving competitive performance compared to other methods (see fig. 1, each dimension is normalized independently for clear visualization). We further show that by introducing the desired priors, the distilled datasets have the same generalization and transferability as the original ones. Our contributions can be summarized as follows: 1. We prove the priors in the well-trained DMs meet the diversity and generalization requirements of DD. 2. We derive the overlooked representativeness prior from DMs and formalize it into a kernel-induced distance, which guides the sampling dynamic and improves the quality of distilled datasets.

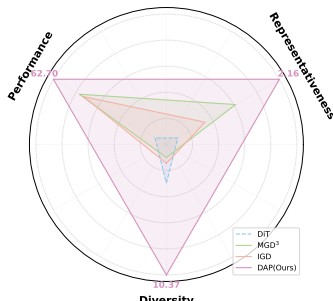

Figure 1: Our diffusion as priors (**DAP**) method is beneficial for the DD task. Diversity: $1+\text{FID}_{max}-\text{FID}$. Representativeness: $\frac{1}{d(\phi(x),\phi(y))}$. Performance: classification results on ImageNet-1K.

## 2 PRELIMINARIES

### 2.1 DATASET DISTILLATION

Given a labeled training dataset $\mathcal{T}_{train} = \{\boldsymbol{x}, \boldsymbol{y}\} \subseteq \mathbb{R}^N \times \mathcal{Y}$ where $\boldsymbol{x} \in \mathbb{R}^N$ i.i.d. drawn from $p_{\text{data}}$, and $\boldsymbol{y} \in \mathcal{Y} = \{1, \ldots, C\}$ drawn from the label space. The objective of DD is to synthesize a compact dataset $\mathcal{S}_{syn} = \{\boldsymbol{x}^{syn}, \boldsymbol{y}\} \subseteq \mathbb{R}^M \times \mathcal{Y}$ $(M \ll N)$ that encapsulates the knowledge of the original data. Consequently, the model trained with small $\mathcal{S}_{syn}$ can achieve considerable generalization performance (measured by loss $\mathcal{L}$) to the large training dataset $\mathcal{T}_{train}$:

$$\mathbb{E}_{\boldsymbol{x}, \boldsymbol{y}, \theta^{(0)}} \left[ \mathcal{L} \left( f_{\text{alg}(\mathcal{T}_{train}, \theta^{(0)})}(\boldsymbol{x}), y \right) \right] \simeq \mathbb{E}_{\boldsymbol{x}, \boldsymbol{y}, \theta^{(0)}} \left[ \mathcal{L} \left( f_{\text{alg}(\mathcal{S}_{syn}, \theta^{(0)})}(\boldsymbol{x}), y \right) \right]. \tag{1}$$

The algorithm $\text{alg}(\cdot, \theta^{(0)})$ is determined by training set $\mathcal{T}$ or $\mathcal{S}$ and the initialized parameters $\theta^{(0)}$.

## 2.2 DIFFUSION MODELS

Given a dataset $\boldsymbol{x}_0 \in \mathbb{R}^N$ i.i.d. drawn from an unknown distribution $q_0(\boldsymbol{x}_0)$, a diffusion model parameterized by $\theta$ tries to learn a distribution $p_\theta(\boldsymbol{x}_0)$ that approximates $q_0(\boldsymbol{x}_0)$. Specifically, the diffusion model places a reversible process that gradually adds Gaussian noise from $\boldsymbol{x}_0$ to $\boldsymbol{x}_T$ at time $T > 0$ and then maps them back. The forward diffusion process is defined by the Itô Stochastic Differential Equation (SDE) Song et al. (2021):

$$\mathrm{d}\boldsymbol{x}_t = f\left(\boldsymbol{x}_t, t\right)\mathrm{d}t + g(t)\mathrm{d}\boldsymbol{w}_t, \tag{2}$$

where $f\left(\boldsymbol{x}_t, t\right) = -\frac{1}{2}\beta_t\boldsymbol{x}_t$ is the drift term and $g(t) = \sqrt{\beta_t}$ denotes the diffusion coefficient that controls the noise strength at each timestep. $\beta_t \in (0, 1)$ is a sequence of pre-defined time-dependent noise scales. Meanwhile, $\boldsymbol{w}_t$ is the Brownian motion. And the reverse diffusion process is given by the time-reverse SDE:

$$\mathrm{d}\boldsymbol{x} = \left[f(\boldsymbol{x}_t, t) - g(t)^2\nabla_{\boldsymbol{x}_t}\log p_t(\boldsymbol{x}_t)\right]\mathrm{d}t + g(t)\mathrm{d}\bar{\boldsymbol{w}}, \tag{3}$$

where $\bar{\boldsymbol{w}}$ represents the time-reversed Brownian motion. The only unknown term in eq. (3) is the *score function* $\nabla_{\boldsymbol{x}_t}\log p_t(\cdot)$ of distribution $p_t$ at each time $t$ (we use $p$ for simplicity). A neural network $\boldsymbol{\epsilon}_\theta(\boldsymbol{x}_t, t)$ is trained to estimate the *score function* $-\nabla_{\boldsymbol{x}_t}\log p(\boldsymbol{x}_t)$. Finally, we can sample $\boldsymbol{x}_0$ by solving the reverse diffusion SDE (Lu et al., 2022).

## 2.3 INHERENT PRIORS IN DIFFUSION MODELS

A key question in evaluating generative models is whether they capture the full variability of the dataset (Alaa et al., 2022). DMs inherently encode diversity and generalization priors through estimating $\nabla_{\boldsymbol{x}}\log p(\boldsymbol{x})$, which compels the model to capture global manifold geometry rather than memorizing individual samples, thereby avoiding mode collapse (Thanh-Tung & Tran, 2020). Moreover, the stochastic perturbations in the forward process act as implicit regularizers, enforcing Lipschitz continuity and improving robustness to distributional shifts (Chen et al., 2024a;b).

In addition, let $H$ denote entropy, and $\varphi$ is an inception classifier. High Inception Score (IS) indicates uniform class coverage (high $H(p_\varphi(y))$) and discriminative sample quality (low $H(p_\varphi(y|\boldsymbol{x}))$). While low Fréchet Inception Distance (FID) certifies alignment between generated and real distributions ($p_{syn} \simeq p_{\mathrm{data}}$). Empirical results (Dhariwal & Nichol, 2021) demonstrate that the structure-induced priors within DMs produce sufficient diversity and generalization.

## 3 DIFFUSION AS PRIORS

### 3.1 MOTIVATION

An ideal distilled dataset should satisfy (Gu et al., 2024; Su et al., 2024):

Distilled Dataset s.t. Diversity $+$ Generalization $+$ Representativeness.

These attributes enable the distilled dataset to be effectively applied across a variety of tasks, yielding competitive performance. ***Diversity*** ensures that synthetic data captures the full variability present in the original data, while ***Generalization*** prevents overfitting to the architecture of distillation models. Most importantly, ***Representativeness*** guarantees that the synthetic data retains the most critical information from the raw dataset. Consequently, we seek to study: *how to align the priors of DMs with these attributes and make the distilled dataset desirable?*

Formally, the objective of DMs that estimates the score function $\nabla_{\boldsymbol{x}}\log p(\boldsymbol{x})$ provides synthetic dataset with inherent diversity and generalization priors (discussed in section 3.2). In terms of the representativeness prior $\mathcal{R}$, we consider introducing it into the score function as a condition. According to Bayes' theorem, the conditional score function can be decomposed as:

$$\nabla_{\boldsymbol{x}}\log p(\boldsymbol{x}|\mathcal{R}) = \underbrace{\nabla_{\boldsymbol{x}}\log p(\boldsymbol{x})}_{\text{Diversity \& Generalization}} + \underbrace{\nabla_{\boldsymbol{x}}\log p(\mathcal{R}|\boldsymbol{x})}_{\text{Representativeness}}. \tag{4}$$

Given a well-trained diffusion model, the first term in eq. (4), same as the original score function, is already estimated by $\boldsymbol{\epsilon}_\theta$. Thus, we focus on the second term to fulfill the representativeness requirement during sampling (discussed in section 3.3).

## 3.2 Diffusion As Diversity And Generalization Priors

In the field of DD, diversity is characterized by the breadth of feature distribution and comprehensive coverage of categorical information. Meanwhile, generalization refers to the ability to prevent overfitting to the training data and enable datasets with cross-architecture adaptation. These properties enable the distilled dataset to reflect the information and knowledge of the original dataset like a mirror. In this section, we argue that *the pre-trained diffusion model provides inherent diversity and generalization priors for dataset distillation.*

### 3.2.1 Inherent Diversity And Generalization Priors

As mentioned in section 2.3, diffusion models provide a principled foundation for DD, since effective DD requires distilled data that both cover diverse modes (diversity) and faithfully approximate the original dataset distribution (generalization). We quantify these properties with likelihood-based evaluations. The negative log-likelihood (NLL) is defined as $\mathcal{L}_{\text{NLL}} = -\mathbb{E}_{\boldsymbol{x} \sim p_{\text{data}}}[\log p_\theta(\boldsymbol{x})]$. Identical and low NLL values on training and testing sets indicate that $p_\theta(\boldsymbol{x})$ converges to $p_{\text{data}}$ instead of overfitting (see table 1).

Table 1: NLLs ↓ on different datasets. The results are computed by a vanilla diffusion model (Ho et al., 2020) trained on ImageNet.

| Dataset | Training Set | Test Set |
|---|---|---|
| ImageNette | $2.4452_{\pm 1.03}$ | $2.6327_{\pm 1.08}$ |
| ImageWoof | $2.5856_{\pm 0.88}$ | $3.0838_{\pm 0.86}$ |

### 3.2.2 Beyond Prior: Cross-Architecture Generalization

Unlike conventional DD methods that match training dynamics (e.g., Gradients, Parameters, and features) of specific downstream classifiers, DMs distill datasets without pixel-level optimization. The distilled dataset captures data-relevant rather than architecture-relevant knowledge, eliminating dependence on predefined classifier architectures. This architecture-agnostic DD paradigm produces distilled datasets with cross-architecture generalization, enhancing their versatility.

## 3.3 Diffusion As Representativeness Prior

Representative samples refer to a subset of data that accurately reflects the characteristics of the larger population from which it is drawn (Gabbay et al., 2011). Generating a more representative dataset leads to better dataset distillation performance. In this section, we argue that *a well-trained diffusion model itself can serve as a representativeness prior.*

### 3.3.1 representativeness prior in DMs

To capture the *representativeness* prior hidden in the DMs backbone network, we require a similarity measure that quantifies how closely a synthetic sample reflects the characteristics of the real sample. Kernel function is a simple yet effective tool for defining similarity, allowing us to a) express *representativeness* through an induced distance and b) inject this *representativeness* as a differentiable energy term into the sampling process. Formally, let kernel function $\mathcal{K}(x, y) : \mathcal{X} \times \mathcal{X} \to \mathbb{R}$ be the smooth and differentiable similarity measurement which characterizes the similarity between a synthetic sample $x^{syn}$ and a single training sample $x^{train}$. We argue that the larger the similarity between the synthetic samples and the entire training set $\mathbb{E}_{\boldsymbol{x}^{train}}[\mathcal{K}(\boldsymbol{x}^{syn}, \boldsymbol{x}^{train})]$, the better *representativeness* of $\boldsymbol{x}^{syn}$ to the raw dataset. Suppose that $\mathcal{D}_{\mathcal{K}}(x, y)$ is a distance measure induced by the kernel function $\mathcal{K}$. Typically, we expect $\mathcal{D}_{\mathcal{K}}$ to satisfy the fundamental properties of the distance measures. The following theorem demonstrates that, as long as the kernel function $\mathcal{K}$ is positive semi-definite (PSD), the induced distance $\mathcal{D}_{\mathcal{K}}$ is a well-defined distance measure.

**Theorem 3.1.** *Let $\mathcal{K} : \mathcal{X} \times \mathcal{X} \to \mathbb{R}$ be a PSD kernel. Then the $\mathcal{K}$-induced distance measure*

$$\mathcal{D}_{\mathcal{K}}(x, y) = \left[ \mathcal{K}(x, x) + \mathcal{K}(y, y) - 2\mathcal{K}(x, y) \right]^{1/2} \tag{5}$$

*satisfies:*

1. **Non-negativity:** $\mathcal{D}_{\mathcal{K}}(x, y) \geq 0$, *and* $\mathcal{D}_{\mathcal{K}}(x, y) = 0$ *if and only if* $x = y$.

2. **Symmetry:** $\mathcal{D}_{\mathcal{K}}(x, y) = \mathcal{D}_{\mathcal{K}}(y, x)$.

3. **Triangle inequality:** *For any $x, y, z \in \mathcal{X}$, $\mathcal{D}_{\mathcal{K}}(x, z) + \mathcal{D}_{\mathcal{K}}(z, y) \geq \mathcal{D}_{\mathcal{K}}(x, y)$.*

*Proof.* (Sketch, details in section A.2.1) According to Mercer's theorem (Mercer, 1909), the distance metric induced by the PSD kernel can be expressed as the Hilbert norm in reproducing kernel Hilbert space (RKHS), which satisfies the property of norms. □

Therefore, $\mathcal{D}_{\mathcal{K}}$ is a valid distance metric. The Mercer kernel $\mathcal{K}_{\mathcal{M}}$ is a family of PSD kernels that guarantees the existence of a spectral expansion under continuity and compact conditions. Thanks to these desirable properties, we adopt Mercer kernel as the representativeness measure in our method.

**Theorem 3.2.** *Let $\mathcal{K} : \mathcal{X} \times \mathcal{X} \to \mathbb{R}$ be a Mercer kernel, then the $\mathcal{K}$-induced distance $\mathcal{D}_{\mathcal{K}}$ can be factorized as $\mathcal{D}_{\mathcal{K}}(x, y) = d \circ (\phi \times \phi)(x, y)$, where $\phi$ is a feature mapping and $d$ is a simple norm in Hilbert space.*

*Proof.* (Sketch, details in section A.2.2) According to the reproducing property of the kernel function, there exists a mapping $\Phi$ and a feature space $\mathcal{H}$ that allows the kernel $\mathcal{K}$ to be factorized into $\mathcal{K} = \langle \Phi(\cdot), \Phi(\cdot) \rangle_{\mathcal{H}_{\mathcal{K}}}$. The distance formalized by the linear combination of kernel functions can then be factorized into a combination of the complex $\Phi$ and a simple norm $\| \cdot \|_{\mathcal{H}_{\mathcal{K}}}$. □

Mercer kernel allows us to quantify representativeness in RKHS, and the associated kernel-induced measure ensures the underlying optimization problem remains convex and tractable. Hence, the task reduces to identifying a suitable feature extractor $\phi$ that maps inputs into feature space, where the distance metric $d\left(\phi\left(x\right), \phi\left(y\right)\right) \propto \frac{1}{\mathcal{K}(x,y)}$. We posit that the diffusion model itself is a good feature extractor, supported by two observations: its strong image-text alignment reflects a comprehensive understanding of visual content (Yang & Wang, 2023), and its performance as a discriminative classifier exhibits high accuracy, robustness, and certified robustness (Chen et al., 2024a;b).

We propose **D**iffusion **A**s **P**riors (DAP), which utilizes the diversity, generalization, and representativeness priors contained in the well-trained diffusion models to distill datasets. Specifically, the backbone networks (e.g., U-Net or Transformer) are viewed as a mapping function $\phi : \mathcal{X} \to \mathbb{R}^n$, transforming an image $x$ or latent code $z$ into an $n$-dimensional feature vector. During the pre-training phase, the backbones are endowed with the *representativeness* prior, which enables them to capture meaningful and high-level features.

### 3.3.2 GUIDANCE OF REPRESENTATIVENESS PRIOR

We formalize the conditional probability of representativeness term in eq. (4) as a Boltzmann distribution w.r.t. $\mathcal{D}_{\mathcal{K}}$:

$$p(\mathcal{R}|\boldsymbol{x}^{syn}) \triangleq \frac{\{\exp\left[-\frac{1}{N}\sum_N \mathcal{D}_{\mathcal{K}}\left(\boldsymbol{x}^{syn}, \boldsymbol{x}^{train}\right)\right]\}^{\gamma}}{Z}, \tag{6}$$

where $Z > 0$ denotes the normalizing constant, and $\gamma > 0$ controls the scale of representativeness prior. According to theorem 3.2, the conditional score function of representativeness term is:

$$
\begin{aligned}
\nabla_{\boldsymbol{x}^{syn}} \log p(\mathcal{R}|\boldsymbol{x}^{syn}) &= \nabla_{\boldsymbol{x}^{syn}} \log \frac{\{\exp\left[-\frac{1}{N}\sum_N \mathcal{D}_{\mathcal{K}}\left(\boldsymbol{x}^{syn}, \boldsymbol{x}^{train}\right)\right]\}^{\gamma}}{Z} \\
&= \nabla_{\boldsymbol{x}^{syn}} \log \frac{\{\exp\left[-\frac{1}{N}\sum_N d\left(\phi(\boldsymbol{x}^{syn}), \phi(\boldsymbol{x}^{train})\right)\right]\}^{\gamma}}{Z} \\
&\propto -\gamma \frac{1}{N} \sum_N \nabla_{\boldsymbol{x}^{syn}} d\left(\phi(\boldsymbol{x}^{syn}), \phi(\boldsymbol{x}^{train})\right),
\end{aligned} \tag{7}
$$

which is referred to as energy-based guidance (Dhariwal & Nichol, 2021). Practically, we use the classifier guidance method, which employs the pre-trained diffusion itself as a training-free time-dependent classifier $\phi(x_t)$ such that $\phi(x_t, t) \approx \phi(x_0)$ (Shen et al., 2024). Therefore, the reverse diffusion process with guidance is defined as:

$$
\begin{aligned}
\mathrm{d}\boldsymbol{x} &= \left[f(\boldsymbol{x}_t^{syn}, t) - g(t)^2(\nabla_{\boldsymbol{x}_t^{syn}} \log p(\boldsymbol{x}_t^{syn}) + \gamma \nabla_{\boldsymbol{x}_t^{syn}} \log p(\mathcal{R}|\boldsymbol{x}_t^{syn}))\right] \mathrm{d}t + g(t)\mathrm{d}\bar{\boldsymbol{w}} \\
&\propto \left[f(\boldsymbol{x}_t^{syn}, t) - g(t)^2\left(-\boldsymbol{\epsilon}_\theta\left(\boldsymbol{x}_t^{syn}, t\right) + \gamma \nabla_{\boldsymbol{x}_t^{syn}} d\left(\phi\left(\boldsymbol{x}_t^{syn}\right), \phi\left(\boldsymbol{x}_t^{train}\right)\right)\right)\right] \mathrm{d}t + g(t)\mathrm{d}\bar{\boldsymbol{w}},
\end{aligned} \tag{8}
$$

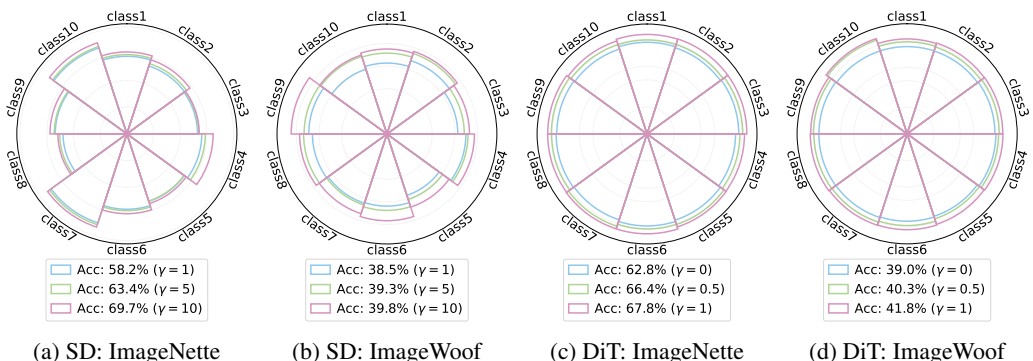

| (a) SD: ImageNette | (b) SD: ImageWoof | (c) DiT: ImageNette | (d) DiT: ImageWoof |

Figure 2: Visualization of average *representativeness* ($\propto \frac{1}{d(\phi(x),\phi(y))}$) of distilled samples (IPC10). As $\gamma$ increases, the representativeness (sector area) gets larger, yielding better DD performance.

where $\nabla_{\boldsymbol{x}_t^{syn}} \log p(\mathcal{R}|\boldsymbol{x}_t^{syn})$ is treated as an auxiliary score derived from the *representativeness* prior. $\boldsymbol{x}_t^{syn}$ and $\boldsymbol{x}_t^{train}$ are the noised $\boldsymbol{x}^{syn}$ and $\boldsymbol{x}^{train}$ at timestep t.

Empirically, we compare the salient features across samples using the linear kernel (Mercer kernel $\mathcal{K}(x, y) = x^\top y$) due to its tractability. As indicated by eq. (6), the representativeness of $\boldsymbol{x}^{syn}$ increases as the energy $\mathcal{D}_\mathcal{K}$ decreases. Figure 2 visualizes the representativeness of class-wise samples under different setups. The distillation performance improves on synthetic samples with higher representativeness, as reflected by the area of the sector. It is worth noting that according to eq. (4), the gradient field of diversity and generalization ($\nabla_{\boldsymbol{x}} \log p(\boldsymbol{x})$) is determined and fixed by pre-trained DMs. Therefore, the gradient field of representativeness cannot be increased indefinitely, otherwise the other priors will lose their effectiveness (see section 4.4 and section A.4.1).

Hereto, we successfully distilled the priors within DMs into the synthetic dataset. Specifically, Diversity prior arises from the stochasticity of diffusion trajectories where different noise initializations lead to distinct denoising paths. Generalization prior stems from the original score function $\nabla_{\boldsymbol{x}} \log p(\boldsymbol{x})$ estimated by vanilla DM. Representativeness prior is directly implemented by the guidance term in eq. (8), which guides each denosing trajectory toward gradient regions that are well represented by real data. We implement the guided sampling process using VP-SDE and summarize the procedure in algorithm 1. The extensive experimental results in section 4 demonstrate the validity of our "Diffusion As Priors (DAP)" method.

---

**Algorithm 1** DAP Sampling (VP-SDE)

---

**Require:** Noisy data samples $\boldsymbol{x}_t^{train|c}$ within class $c$, pre-trained diffusion model $\boldsymbol{\epsilon}_\theta$, a layer output $\phi$ from diffusion backbone network, a Mercer Kernel induced distance measurement $d$, energy-based guidance scale $\gamma$, pre-defined noise scales $\beta_t$.
1: $\boldsymbol{x}_T \sim \mathcal{N}(0, I)$
2: **for** $t = T, \cdots 1$ **do**
3:     $\boldsymbol{\epsilon} \sim \mathcal{N}(0, I)$ **if** $t > 1$, **else** $\boldsymbol{\epsilon} = \mathbf{0}$
4:     $\tilde{\boldsymbol{x}}_{t-1} = (2 - \sqrt{1 - \beta_t})\boldsymbol{x}_t + \beta_t \boldsymbol{\epsilon}_\theta(\boldsymbol{x}_t, t) + \sqrt{\beta_t}\boldsymbol{\epsilon}$
5:     $\boldsymbol{z}_t = \phi(\boldsymbol{x}_t), \boldsymbol{z}_t^{train|c} = \phi(\boldsymbol{x}_t^{train|c})$          # Diffusion as representativeness priors
6:     $\boldsymbol{g}_t = -\nabla_{\boldsymbol{x}_t} d(\boldsymbol{z}_t, \boldsymbol{z}_t^{train|c})$
7:     $\boldsymbol{x}_{t-1} = \tilde{\boldsymbol{x}}_{t-1} + \gamma \boldsymbol{g}_t$                        # Guided sampling
8: **end for**
**Output:** $\boldsymbol{x}_0$                                       # The distilled sample of class $c$.

---

# 4 EXPERIMENTS

In this section, we conduct extensive experiments to validate the effectiveness of DAP. Our evaluation aims to explore the following questions:

- Does DAP achieve state-of-the-art performance on large-scale DD benchmarks?
- How do the three priors: diversity, generalization, and representativeness contribute to the effectiveness of DAP?
- Can DAP generalize across network architectures and datasets?

We evaluate DAP on ImageNet-1K and its widely used subsets (ImageNette, ImageWoof, and ImageIDC), comparing against advanced DD methods, including Minimax, $D^4M$, IGD, $MGD^3$, $D^3HR$ and VLCP. We employ two diffusion architectures, U-Net-based Stable Diffusion (SD) and Transformer-based DiT, for distillation. We also use them as baselines to demonstrate the advantage of the diffusion priors. All results are reported under either hard-label or soft-label evaluation protocols, as specified by the benchmarks. Further experimental details are provided in the section 4.1.

## 4.1 EXPERIMENTAL SETUP

### 4.1.1 DATASETS AND BENCHMARKS

We evaluate DAP on a range of benchmarks that vary in scale, resolution, and task difficulty. Our primary evaluation is conducted on large-scale ImageNet-1K ($224 \times 224$) (Deng et al., 2009). To study the effect of inter-class similarity, we further consider two 10-class subsets of ImageNet-1K: ImageNette (Howard, 2019a), which consists of visually distinct categories and represents a relatively simple task, and ImageWoof (Howard, 2019b), which contains visually similar dog breeds and thus poses a fine-grained classification challenge. Additionally, we incorporate ImageIDC (Kim et al., 2022) to evaluate performance.

### 4.1.2 MODELS AND EVALUATION PROTOCOLS

For each dataset, we distill subsets of 10, 50, and 100 images per class (IPC) and assess their utility on downstream classification tasks. Two evaluation protocols are adopted:

- Hard-label protocol: Following Chen et al. (2025), we directly train classifiers from scratch using the distilled images with ground-truth labels (one-hot labels). We evaluate on three commonly used architectures: ConvNet-6, ResNetAP-10, and ResNet-18.
- Soft-label protocol: Following Sun et al. (2024), we provide soft labels via pre-trained classifiers (e.g., ResNet-18). This protocol is crucial for challenging datasets such as ImageNet-1K, where training from scratch on a few synthetic images is relatively difficult.

To demonstrate the compatibility of DAP, we conduct experiments on a) Stable Diffusion-V1.5 with the U-Net backbone, and b) DiT-XL/2-256 with the transformer.

### 4.1.3 OTHER DETAILS

All experiments were implemented in PyTorch and conducted with NVIDIA A40 GPUs. For fair comparison, we reproduce baseline methods under the same setup. The reported results follow these conventions: a) For DAP and reproduced baselines, we report the mean$_{\pm\text{standard deviation}}$ over three runs. b) For other methods, we report results from the original papers. c) In tables, the **best** result is highlighted in bold, while the second best is underlined. Practically, DAP does not require selecting any specific $x^{train|c}$ samples before sampling.

## 4.2 COMPARISON WITH STATE-OF-THE-ART METHODS

### 4.2.1 RESULTS ON DiT

We begin with ImageNet-1K, the most widely adopted benchmark for generative dataset distillation. Across both IPC (Images Per Class) settings, DAP consistently achieves the best results, demonstrating its superiority in large-scale DD tasks. As listed in table 2, DAP achieves $49.1\%$ Top-1 accuracy at IPC10, exceeding the strongest baseline IGD and $MGD^3$ by 3.5%. With more distilled samples, DAP further improves to $62.7\%$, establishing superior results on this challenging benchmark.

To examine robustness across different scales and architectures, we also evaluate on ImageNet subsets, including ImageNette and ImageWoof (table 3). DAP again outperforms almost all competing

Table 2: Top-1 Accuracy on **ImageNet-1K**. The results are evaluated with **soft-label protocol** based on ResNet-18.

| Dataset | IPC | SRe$^2$L | G-VBSM | RDED | Minimax | DiT | IGD | MGD$^3$ | D$^3$HR | VLCP | DAP |
|---|---|---|---|---|---|---|---|---|---|---|---|
| ImageNet-1K | 10 | $21.3_{\pm0.6}$ | $31.4_{\pm0.5}$ | $42.0_{\pm0.1}$ | $44.3_{\pm0.5}$ | $39.6_{\pm0.4}$ | $45.5_{\pm0.5}$ | $45.6_{\pm0.8}$ | $44.3_{\pm0.3}$ | $\underline{46.7}_{\pm0.4}$ | $\mathbf{49.1}_{\pm1.2}$ |
| | 50 | $46.8_{\pm0.2}$ | $51.8_{\pm0.4}$ | $56.5_{\pm0.1}$ | $58.6_{\pm0.3}$ | $52.9_{\pm0.6}$ | $59.8_{\pm0.3}$ | $60.2_{\pm0.1}$ | $59.4_{\pm0.1}$ | $\underline{60.5}_{\pm0.2}$ | $\mathbf{62.7}_{\pm1.5}$ |

methods. An exception occurs with ResNet-18 at IPC10, IGD slightly surpasses DAP. This deviation is attributed to the fact that IGD explicitly incorporates ResNet-18 as the surrogate network for its influence-guided sampling, thereby introducing an inductive bias favoring the specific architecture. While this bias yields localized gains, it also risks overfitting (see table 5). In contrast, DAP does not rely on architecture-specific heuristics and remains effective across multiple backbones.

Table 3: Top-1 Accuracy on **ImageNette** and **ImageWoof**. The results are evaluated with **hard-label protocol**.

| Dataset | Model | IPC | Random | DM | DiT | Minimax | IGD | MGD$^3$ | DAP | Full |
|---|---|---|---|---|---|---|---|---|---|---|
| Nette | ConvNet-6 | 10 | $46.0_{\pm0.5}$ | $49.8_{\pm1.1}$ | $56.2_{\pm1.3}$ | $58.2_{\pm0.9}$ | $\underline{61.9}_{\pm1.9}$ | $56.2_{\pm1.7}$ | $\mathbf{64.8}_{\pm0.8}$ | |
| | | 50 | $71.8_{\pm1.2}$ | $70.3_{\pm0.8}$ | $74.1_{\pm0.6}$ | $76.9_{\pm0.9}$ | $\underline{80.9}_{\pm0.9}$ | $79.0_{\pm0.3}$ | $\mathbf{82.2}_{\pm1.6}$ | $94.3_{\pm0.5}$ |
| | | 100 | $79.9_{\pm0.8}$ | $78.5_{\pm0.8}$ | $78.2_{\pm0.3}$ | $81.1_{\pm0.3}$ | $\underline{84.5}_{\pm0.7}$ | $84.4_{\pm0.6}$ | $\mathbf{85.7}_{\pm1.3}$ | |
| | ResNetAP-10 | 10 | $54.2_{\pm1.2}$ | $60.2_{\pm0.7}$ | $62.8_{\pm0.8}$ | $63.2_{\pm1.0}$ | $\underline{66.5}_{\pm1.1}$ | $66.4_{\pm2.4}$ | $\mathbf{67.8}_{\pm1.2}$ | |
| | | 50 | $77.3_{\pm1.0}$ | $76.7_{\pm1.0}$ | $76.9_{\pm0.5}$ | $78.2_{\pm0.7}$ | $\underline{81.0}_{\pm1.2}$ | $79.5_{\pm1.3}$ | $\mathbf{82.3}_{\pm0.7}$ | $94.6_{\pm0.5}$ |
| | | 100 | $81.1_{\pm0.6}$ | $80.9_{\pm0.7}$ | $80.1_{\pm1.1}$ | $81.3_{\pm0.9}$ | $\underline{85.2}_{\pm0.5}$ | $85.0_{\pm0.4}$ | $\mathbf{86.0}_{\pm2.1}$ | |
| | ResNet-18 | 10 | $55.8_{\pm1.0}$ | $60.9_{\pm0.7}$ | $62.5_{\pm0.9}$ | $64.9_{\pm0.6}$ | $\mathbf{67.7}_{\pm0.3}$ | $61.2_{\pm1.4}$ | $\underline{66.4}_{\pm0.5}$ | |
| | | 50 | $75.8_{\pm1.1}$ | $75.0_{\pm1.0}$ | $75.2_{\pm0.9}$ | $78.1_{\pm0.6}$ | $\underline{81.0}_{\pm0.7}$ | $80.8_{\pm0.9}$ | $\mathbf{82.8}_{\pm1.1}$ | $95.3_{\pm0.6}$ |
| | | 100 | $82.0_{\pm0.4}$ | $81.5_{\pm0.4}$ | $77.8_{\pm0.6}$ | $81.3_{\pm0.7}$ | $\underline{84.4}_{\pm0.8}$ | $83.7_{\pm1.3}$ | $\mathbf{85.5}_{\pm1.5}$ | |
| Woof | ConvNet-6 | 10 | $25.2_{\pm1.1}$ | $27.6_{\pm1.2}$ | $32.3_{\pm0.8}$ | $33.5_{\pm1.4}$ | $\underline{35.0}_{\pm0.8}$ | $34.7_{\pm1.1}$ | $\mathbf{37.6}_{\pm0.9}$ | |
| | | 50 | $41.9_{\pm1.4}$ | $43.8_{\pm1.1}$ | $48.5_{\pm1.3}$ | $50.7_{\pm1.8}$ | $54.2_{\pm0.7}$ | $\underline{54.5}_{\pm1.6}$ | $\mathbf{55.8}_{\pm0.4}$ | $85.9_{\pm0.4}$ |
| | | 100 | $52.3_{\pm1.5}$ | $50.1_{\pm0.9}$ | $54.2_{\pm1.5}$ | $57.1_{\pm1.9}$ | $\underline{61.1}_{\pm1.0}$ | $60.1_{\pm1.2}$ | $\mathbf{62.4}_{\pm1.2}$ | |
| | ResNetAP-10 | 10 | $31.6_{\pm0.8}$ | $29.8_{\pm1.0}$ | $39.0_{\pm0.9}$ | $39.6_{\pm1.2}$ | $\underline{41.0}_{\pm0.8}$ | $40.4_{\pm1.9}$ | $\mathbf{41.8}_{\pm0.7}$ | |
| | | 50 | $50.1_{\pm1.6}$ | $47.8_{\pm1.2}$ | $55.8_{\pm1.1}$ | $59.8_{\pm0.8}$ | $\underline{62.7}_{\pm1.2}$ | $56.5_{\pm1.9}$ | $\mathbf{63.3}_{\pm0.5}$ | $87.2_{\pm0.6}$ |
| | | 100 | $59.2_{\pm0.9}$ | $59.8_{\pm1.3}$ | $62.5_{\pm0.9}$ | $66.8_{\pm1.2}$ | $\underline{69.7}_{\pm0.9}$ | $66.5_{\pm1.0}$ | $\mathbf{70.8}_{\pm1.4}$ | |
| | ResNet-18 | 10 | $30.9_{\pm1.3}$ | $30.2_{\pm0.6}$ | $40.6_{\pm0.6}$ | $42.2_{\pm1.2}$ | $\mathbf{44.8}_{\pm0.8}$ | $38.5_{\pm2.5}$ | $\underline{43.9}_{\pm0.9}$ | |
| | | 50 | $54.0_{\pm0.8}$ | $53.9_{\pm0.7}$ | $57.4_{\pm0.7}$ | $60.5_{\pm0.5}$ | $\underline{62.0}_{\pm1.1}$ | $58.3_{\pm1.4}$ | $\mathbf{63.2}_{\pm0.7}$ | $89.0_{\pm0.6}$ |
| | | 100 | $63.6_{\pm0.5}$ | $64.9_{\pm0.7}$ | $62.3_{\pm0.5}$ | $67.4_{\pm0.7}$ | $\underline{70.6}_{\pm1.8}$ | $68.8_{\pm0.7}$ | $\mathbf{71.6}_{\pm1.3}$ | |

### 4.2.2 RESULTS ON STABLE DIFFUSION

We next apply DAP to Stable Diffusion (SD) as the generative backbone. As shown in fig. 3, DAP consistently surpasses the baseline MGD$^3$ and vanilla Stable Diffusion across all datasets and IPC settings. For instance, DAP reaches 81.4% accuracy on ImageNette with IPC50, approaching the accuracy of training on the full dataset while using only a fraction of the data size.

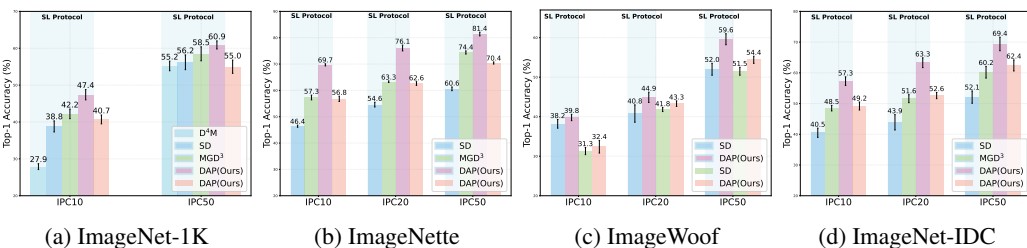

(a) ImageNet-1K      (b) ImageNette      (c) ImageWoof      (d) ImageNet-IDC

Figure 3: The comparison results on **Stable Diffusion**. The results are evaluated with both **hard-label (HL)** and **soft-label (SL) protocols** based on ResNet-18. The results of SL protocol are marked with a light blue background, while those without background color are from HL protocol.

A surprising finding arises when comparing hard-label and soft-label protocols. Most previous methods achieve competitive results only under soft-label supervision, whereas DAP already matches or

surpasses them under the stricter hard-label supervision. This demonstrates that the representativeness prior substantially improves the quality of distilled datasets, even without auxiliary supervision. Moreover, DAP maintains robustness under domain shifts between the SD pre-training dataset (LAION (Schuhmann et al., 2022)) and the distilled dataset (ImageNet), further highlighting its ability to leverage diffusion priors to bridge domain gaps, which is not observed in existing approaches.

## 4.3 ANALYSIS OF DIFFUSION PRIORS

### 4.3.1 GENERALIZATION PRIOR

Many existing methods overfit to the distillation settings and suffer performance degradation when the dataset scale is reduced or the evaluation architecture is changed. As listed in table 4, we obtain IPC50 and IPC10 datasets by subsampling them from IPC100 datasets rather than generating them specially. IGD and MGD[3] suffer degradation under this reduction, whereas DAP preserves accuracy across scales without noticeable performance loss. This generalization indicates that DAP captures sufficient transferable knowledge rather than memorizing samples at a fixed scale.

Table 4: A study on dataset scale reduction. The results are Top-1 Accuracy evaluated with **hard-label protocol**. The failure cases (degradation $> 5\%$ compared to table 3) are marked in blue.

| Model | IPC | ImageNette | | | | ImageWoof | | | |
|---|---|---|---|---|---|---|---|---|---|
| | | IGD | MGD[3] | DAP | Full | IGD | MGD[3] | DAP | Full |
| ConvNet-6 | 10 | $59.8_{\pm2.3}$ | $54.2_{\pm1.9}$ | $\mathbf{64.5}_{\pm0.7}$ | | $32.6_{\pm1.5}$ | $27.0_{\pm1.2}$ | $\mathbf{36.5}_{\pm1.8}$ | |
| | 50 | $79.8_{\pm1.8}$ | $77.0_{\pm1.3}$ | $\mathbf{80.1}_{\pm1.2}$ | $94.3_{\pm0.4}$ | $\mathbf{53.4}_{\pm0.7}$ | $51.4_{\pm0.8}$ | $53.1_{\pm0.9}$ | $85.9_{\pm0.4}$ |
| | 100 | $82.8_{\pm0.6}$ | $83.7_{\pm0.8}$ | $\mathbf{85.7}_{\pm1.3}$ | | $60.2_{\pm0.4}$ | $58.8_{\pm0.8}$ | $\mathbf{62.4}_{\pm1.2}$ | |
| ResNetAP-10 | 10 | $63.2_{\pm1.7}$ | $59.2_{\pm1.6}$ | $\mathbf{66.1}_{\pm0.4}$ | | $35.6_{\pm1.7}$ | $31.8_{\pm1.4}$ | $\mathbf{37.9}_{\pm0.8}$ | |
| | 50 | $73.4_{\pm1.3}$ | $79.0_{\pm1.1}$ | $\mathbf{79.8}_{\pm1.5}$ | $94.6_{\pm0.5}$ | $60.4_{\pm0.7}$ | $58.6_{\pm1.3}$ | $\mathbf{62.6}_{\pm0.6}$ | $87.2_{\pm0.6}$ |
| | 100 | $82.5_{\pm1.2}$ | $83.0_{\pm0.5}$ | $\mathbf{86.0}_{\pm2.1}$ | | $66.8_{\pm0.9}$ | $64.9_{\pm0.4}$ | $\mathbf{70.8}_{\pm1.4}$ | |
| ResNet-18 | 10 | $62.6_{\pm2.1}$ | $56.0_{\pm1.8}$ | $\mathbf{63.7}_{\pm0.8}$ | | $35.2_{\pm1.4}$ | $29.8_{\pm2.3}$ | $\mathbf{39.4}_{\pm1.3}$ | |
| | 50 | $78.4_{\pm1.4}$ | $78.8_{\pm1.6}$ | $\mathbf{80.4}_{\pm2.3}$ | $95.3_{\pm0.6}$ | $59.3_{\pm0.5}$ | $59.4_{\pm1.7}$ | $\mathbf{59.7}_{\pm1.2}$ | $89.0_{\pm0.6}$ |
| | 100 | $83.6_{\pm1.1}$ | $84.2_{\pm0.8}$ | $\mathbf{85.5}_{\pm1.5}$ | | $68.8_{\pm0.8}$ | $67.8_{\pm1.1}$ | $\mathbf{71.6}_{\pm0.9}$ | |

We further evaluate cross-architecture generalization in table 5. The distilled datasets are trained with soft-labels provided by ResNet-18 and tested on other architectures, including ResNet-101, MobileNet-V2, EfficientNet-B0, and Swin Transformer. While baselines show performance drops due to inductive bias on the architectures, DAP consistently achieves the highest accuracy across all cases. These findings confirm that representativeness prior enables architecture-agnostic DD.

Table 5: A study on cross-architecture generalization. The results are Top-1 Accuracy on **ImageNet-1K** evaluated with **soft-label protocol**.

| Method | ResNet-101 | | MobileNet-V2 | | EfficientNet-B0 | | Swin Transformer | |
|---|---|---|---|---|---|---|---|---|
| | IPC10 | IPC50 | IPC10 | IPC50 | IPC10 | IPC50 | IPC10 | IPC50 |
| RDED | $48.3_{\pm1.0}$ | $61.2_{\pm0.4}$ | $40.4_{\pm0.1}$ | $53.3_{\pm0.2}$ | $31.0_{\pm0.1}$ | $58.5_{\pm0.4}$ | $42.3_{\pm0.6}$ | $53.2_{\pm0.8}$ |
| IGD | $52.6_{\pm1.2}$ | $66.2_{\pm0.2}$ | $39.2_{\pm0.2}$ | $57.8_{\pm0.2}$ | $47.7_{\pm0.1}$ | $62.0_{\pm0.1}$ | $44.1_{\pm0.6}$ | $58.6_{\pm0.5}$ |
| DAP | $\mathbf{54.9}_{\pm0.9}$ | $\mathbf{68.1}_{\pm0.4}$ | $\mathbf{43.1}_{\pm0.3}$ | $\mathbf{61.4}_{\pm0.2}$ | $\mathbf{49.7}_{\pm0.3}$ | $\mathbf{65.2}_{\pm0.4}$ | $\mathbf{48.3}_{\pm0.6}$ | $\mathbf{61.7}_{\pm0.4}$ |

### 4.3.2 DIVERSITY AND REPRESENTATIVENESS PRIORS

To investigate whether DAP enforces diversity and representativeness priors in the distilled datasets, we visualize the data distribution using t-SNE alongside both the training and test sets. Figure 4 reveals that the synthetic data aligns well with the training set while generalizing to the test set, demonstrating that the DAP can accurately match the underlying data manifold. Moreover, the embeddings show intra-class diversity and inter-class separability, indicating that the distilled datasets capture meaningful variability without sacrificing discriminability.

Across all benchmarks and analyses, DAP achieves competitive performance and surpasses existing DD methods. The improvements arise from the combined effect of diffusion priors: diversity and

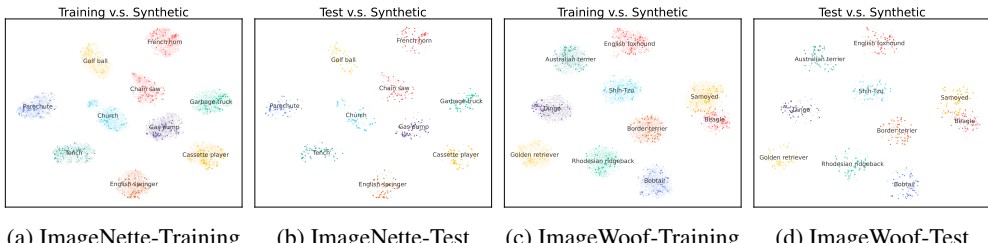

(a) ImageNette-Training    (b) ImageNette-Test    (c) ImageWoof-Training    (d) ImageWoof-Test

Figure 4: Visualization results of t-SNE. We compare the feature distribution of real (training and test set) versus synthetic data under IPC50. Dark/Light points: Synthetic/Real samples.

generalization priors contribute to broad coverage and cross-architecture transfer. Meanwhile, the representativeness prior enforces information alignment with the real dataset. Moreover, DAP introduces no extra training cost, which makes the approach both efficient and scalable in scenarios where deployment architectures are agnostic. We also discuss the sampling costs in section A.3.7.

## 4.4 ABLATION EXPERIMENTS

We conduct ablation studies to investigate the influence of feature layer selection and guidance scale $\gamma$ in representativeness guidance. We observe from fig. 5a that the "Mid" layer of the U-Net yields the strongest results. For DiT, the most effective features originate from the early transformer blocks (e.g., the 4th-12th layers shown in fig. 5b), which outperform those in later layers. Despite this difference, both cases consistently reveal that the final output layers are suboptimal for representativeness guidance, as they prioritize distribution alignment over representativeness. Regarding $\gamma$, we find that increasing its value generally enhances representativeness, as reflected by improved downstream accuracy in figs. 5c and 5d and the sector areas in fig. 2, but excessive scales distort the gradient field of the sampling process and bias the generation trajectory, thereby diminishing the contributions of diversity and generalization priors and leading to performance degradation.

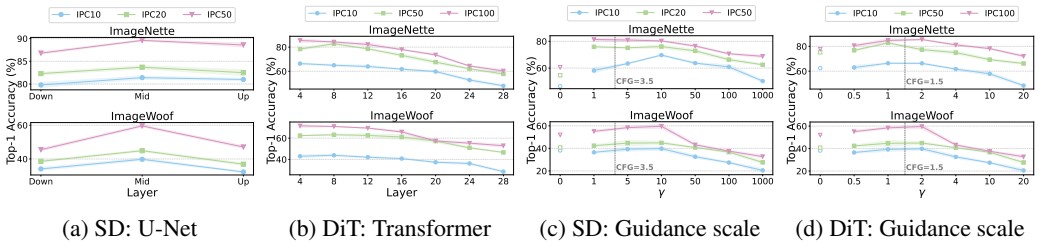

(a) SD: U-Net    (b) DiT: Transformer    (c) SD: Guidance scale    (d) DiT: Guidance scale

Figure 5: Ablation studies under **ResNet-18**. (a-b) Top-1 Accuracy under different backbone layer selection. (c-d) Top-1 Accuracy under varied guidance scale $\gamma$.

## 5 CONCLUSION

This paper introduces Diffusion as Priors, a framework for dataset distillation that leverages the inherent priors of diffusion models. We identify diversity, generalization, and representativeness priors in diffusion models, and demonstrate how they can be integrated to guide the generation process. Representativeness prior is further formulated through kernel–based energy guidance, enabling the sampling process to align more information with real data. Extensive experiments on ImageNet-1K and its subsets demonstrated that DAP achieves state-of-the-art results, preserves generalization under scale reduction, transfers effectively across architectures, and remains robust under domain shifts, making the approach both efficient and scalable. Future work may fall in extending diffusion priors to other powerful models (e.g., FLUX, Stable Diffusion 3.5) and exploring applications beyond vision, including language, video, and multimodal datasets.

ETHICS STATEMENT

This work uses only publicly available datasets, including ImageNet-1K and its subsets (ImageNette, ImageWoof, and ImageIDC). No human subjects, private data, or sensitive information are involved.

REPRODUCIBILITY STATEMENT

We have taken several measures to ensure reproducibility. All theoretical results are presented with complete proofs in section A.2. Details of datasets, backbone architectures, hyper-parameters, and evaluation protocols are provided in section 4.1, while algorithms 1 and 2 specify the guided sampling procedure. Additional visualizations and ablation results are included in section A.4 to further support empirical findings. These resources ensure that all theoretical and experimental results can be independently verified.

ACKNOWLEDGMENTS

This work was supported by the Fundamental and Interdisciplinary Disciplines Breakthrough Plan of the Ministry of Education of China (No. JYB2025XDXM101), CPSF (No. 2025M771546) and NSFC (No. 625B200186).

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

# A    APPENDIX

Appendix organization:

## A.1    BACKGROUND

### A.1.1    DATASET DISTILLATION

Systematic analysis of research in dataset distillation reveals two paradigms: a) traditional matching-based approaches focused on pixel-level optimization, and b) modern generative frameworks emphasizing distribution learning (Yu et al., 2023; Lei & Tao, 2023; Liu & Du, 2025). Traditional methods adopt an "imitation" philosophy, involving continuous pixel optimization to align model behavior, such as gradients, feature distributions, or checkpoints between synthetic and original data (Zhao & Bilen, 2021; Wang et al., 2022; Zhao et al., 2023; Deng et al., 2024). In contrast, generative frameworks prioritize improving dataset quality through fidelity and diversity metrics. These approaches extract key informational patterns from source data, enhancing the realism and generalization of distilled datasets. We will examine the related work in the following subsection.

### A.1.2    GENERATIVE DATASET DISTILLATION

Generative dataset distillation utilizes models such as Generative Adversarial Networks (GANs) and Diffusion models (DMs) to synthesize compact and informative datasets. Unlike pixel optimization methods, which are limited to small-scale, low-resolution data due to computational costs, generative techniques support large-scale, high-resolution applications. This flexibility promotes sample diversity and better generalization across model architectures. This section reviews the two primary categories of generative dataset distillation methods: GAN-based and Diffusion-based approaches.

**GAN-based approaches.**    GANs serve as foundation models for dataset distillation In early research. DiM (Wang et al., 2023) condenses dataset information into a conditional GAN, enabling sample synthesis from random noise during deployment. GLaD (Cazenavette et al., 2023) enhances cross-architecture generalization by distilling data into the latent space of pre-trained models like StyleGAN (Karras et al., 2019). H-PD (Zhong et al., 2024) introduces hierarchical parameterization distillation, optimizing across latent spaces in GANs to capture hierarchical features from the initial latent space to the pixel space.

**Diffusion-based approaches.** Diffusion-based methods leverage diffusion models to improve dataset distillation. For example, Minimax diffusion (Gu et al., 2024) fine-tunes a diffusion model with minimax criteria to boost representativeness and diversity. VLCP (Zou et al., 2025) constructs text prototypes to enrich the labels with semantic information and then fine-tunes the DMs with image-text pairs. $D^4M$ (Su et al., 2024) disentangles feature extraction and generation via Training-Time Matching (TTM) with category prototypes. IGD (Chen et al., 2025) guides the sampling process of pre-trained diffusion models using a function combining trajectory influence and diversity constraints, generating synthetic data without retraining. Additionally, $MGD^3$ (Santiago et al., 2025) enhances diversity by identifying latent space modes and directing data toward them during sampling. In order to enhance the objective and conditional consistency of the distillation process, $CaO_2$ (Wang et al., 2025) employs target-guided sample selection to optimize the latent conditionally. $D^3HR$ (Zhao et al., 2025) utilizes DDIM inversion to map the image latents to the Gaussian domain, then aligns the representative latents with the high-normality Gaussian distribution with their proposed sampling scheme.

## A.2 PROOFS

### A.2.1 VALIDITY OF KERNEL-INDUCED DISTANCE

**Theorem A.1.** *Let* $\mathcal{K} : \mathcal{X} \times \mathcal{X} \to \mathbb{R}$ *be a PSD kernel. Then the induced distance measure*

$$\mathcal{D}_\mathcal{K}(x,y) = \left[\mathcal{K}(x,x) + \mathcal{K}(y,y) - 2\mathcal{K}(x,y)\right]^{1/2} \tag{9}$$

*satisfies:*

1. **Non-negativity:** $\mathcal{D}_\mathcal{K}(x,y) \geq 0$, *and* $\mathcal{D}_\mathcal{K}(x,y) = 0$ *if and only if* $x = y$.

2. **Symmetry:** $\mathcal{D}_\mathcal{K}(x,y) = \mathcal{D}_\mathcal{K}(y,x)$.

3. **Triangle inequality:** *For any* $x, y, z \in \mathcal{X}$, $\mathcal{D}_\mathcal{K}(x,z) + \mathcal{D}_\mathcal{K}(z,y) \geq \mathcal{D}_\mathcal{K}(x,y)$.

*Proof.* Since $\mathcal{K}$ is positive semi-definite, by Mercer's theorem (Mercer, 1909) there exists a reproducing kernel Hilbert space $\mathcal{H}$ and a feature map $\phi : \mathcal{X} \to \mathcal{H}$ such that

$$\mathcal{K}(x,y) = \langle \phi(x), \phi(y) \rangle_\mathcal{H}. \tag{10}$$

Therefore,

$$\mathcal{D}_\mathcal{K}(x,y)^2 = \mathcal{K}(x,x) + \mathcal{K}(y,y) - 2\mathcal{K}(x,y) \tag{11}$$
$$= \langle \phi(x), \phi(x) \rangle_\mathcal{H} + \langle \phi(y), \phi(y) \rangle_\mathcal{H} - 2\langle \phi(x), \phi(y) \rangle_\mathcal{H} \tag{12}$$
$$= \|\phi(x) - \phi(y)\|_\mathcal{H}^2. \tag{13}$$

Thus,

$$\mathcal{D}_\mathcal{K}(x,y) = \|\phi(x) - \phi(y)\|_\mathcal{H}. \tag{14}$$

Since the norm in Hilbert space $\|\cdot\|_\mathcal{H}$ is a valid metric, it satisfies:

- Non-negativity and identity of indiscernibles: $\|\phi(x) - \phi(y)\| \geq 0$, and $\|\phi(x) - \phi(y)\| = 0$ iff $\phi(x) = \phi(y)$, which implies $x = y$.

- Symmetry: $\|\phi(x) - \phi(y)\| = \|\phi(y) - \phi(x)\|$.

- Triangle inequality: $\|\phi(x) - \phi(y)\| \leq \|\phi(x) - \phi(z)\| + \|\phi(z) - \phi(y)\|$ for any $z$.

Therefore, $\mathcal{D}_\mathcal{K}$ is a valid metric induced by the kernel $\mathcal{K}$. □

### A.2.2 DISTANCE FACTORIZATION

**Theorem A.2.** *Let* $\mathcal{K} : \mathcal{X} \times \mathcal{X} \to \mathbb{R}$ *be a Mercer kernel, then the induced chordal distance* $\mathcal{D}_\mathcal{K}$ *can be factorized as* $\mathcal{D}_\mathcal{K}(x,y) = d \circ (\phi \times \phi)(x,y)$, *where* $\phi$ *is a feature mapping and* $d$ *is a simple norm in Hilbert space.*

*Proof.* By the reproducing property of the reproducing kernel Hilbert space $\mathcal{H}_{\mathcal{K}}$, we have

$$\mathcal{K}(x,y) = \langle \Phi(x), \Phi(y) \rangle_{\mathcal{H}_{\mathcal{K}}}. \tag{15}$$

Therefore,

$$\mathcal{D}_{\mathcal{K}}(x,y)^2 = \mathcal{K}(x,x) + \mathcal{K}(y,y) - 2\mathcal{K}(x,y) \tag{16}$$

$$= \langle \Phi(x), \Phi(x) \rangle_{\mathcal{H}_{\mathcal{K}}} + \langle \Phi(y), \Phi(y) \rangle_{\mathcal{H}_{\mathcal{K}}} - 2\langle \Phi(x), \Phi(y) \rangle_{\mathcal{H}_{\mathcal{K}}} \tag{17}$$

$$= \|\Phi(x) - \Phi(y)\|_{\mathcal{H}_{\mathcal{K}}}^2. \tag{18}$$

Taking square roots yields

$$\mathcal{D}_{\mathcal{K}}(x,y) = \|\Phi(x) - \Phi(y)\|_{\mathcal{H}_{\mathcal{K}}}. \tag{19}$$

If we set $f = \Phi$ and $d(u,v) = \|u - v\|_{\mathcal{H}_{\mathcal{K}}}$, then clearly

$$\mathcal{D}_{\mathcal{K}}(x,y) = d(f(x), f(y)) = d \circ (f \times f)(x,y). \tag{20}$$

$\square$

## A.3 DISCUSSIONS

### A.3.1 COMPATIBILITY OF DAP

**Compatibility with codebase.** DAP is fully implemented using the native components of the diffusion model itself, without relying on any additional modules or external dependencies. This carefully designed approach not only preserves complete compatibility with existing diffusion architectures but also ensures that the method can be readily adopted across diverse codebases. As a result, DAP can be seamlessly incorporated into widely used diffusion libraries, such as the `Diffusers` library in Hugging Face, thereby promoting both reproducibility and broad applicability in contemporary research and practical deployment scenarios.

**Compatibility with other methods.** Since the DAP pipeline works orthogonally with the existing approaches, it exhibits strong compatibility, allowing it to complement them without interference. Before empirical results, we analyze the sources of representativeness priors employed by different methods in table 6: Minimax introduces representativeness via fine-tuning under the supervision of the proposed training loss, $D^4M$ and $MGD^3$ capture representativeness through clustering algorithms, IGD and $CaO_2$ uses pre-trained classifier, $D^3HR$ calculates the key statistics including mean, standard deviation and skewness, while DAP exploits the representativeness priors embedded in diffusion models. To validate the compatibility of DAP, we incorporate it into Minimax for instance

Table 6: The source of representativeness knowledge from different generative DD methods.

| Method | Representativeness |
|---|---|
| Minimax | Training Loss |
| $D^4M$ | Clustering |
| $MGD^3$ | Clustering |
| IGD | Pre-trained Classifier |
| $CaO_2$ | Pre-trained Classifier |
| $D^3HR$ | Statistic Metric |
| DAP | Diffusion Prior |

(see table 7). The addition of DAP consistently enhances the performance of distilled samples. These results indicate that DAP can serve as a versatile and modular enhancement, improving the performance of DD approaches while preserving its intrinsic advantages.

Table 7: Top-1 Accuracy on **ImageNette** and **ImageWoof**. Evaluated with **hard-label protocol**.

| Model | IPC | ImageNette | | | ImageWoof | | |
|---|---|---|---|---|---|---|---|
| | | Minimax | Minimax-IGD | Minimax-DAP | Minimax | Minimax-IGD | Minimax-DAP |
| ConvNet-6 | 10 | $58.2_{\pm0.9}$ | $\underline{58.8_{\pm1.0}}$ | $\mathbf{64.2_{\pm1.4}}$ | $33.5_{\pm1.4}$ | $\underline{36.2_{\pm1.6}}$ | $\mathbf{38.2_{\pm0.8}}$ |
| | 50 | $76.9_{\pm0.9}$ | $\underline{82.3_{\pm0.8}}$ | $\mathbf{83.5_{\pm0.6}}$ | $50.7_{\pm1.8}$ | $\underline{55.7_{\pm0.8}}$ | $\mathbf{55.9_{\pm1.2}}$ |
| ResNetAP-10 | 10 | $63.2_{\pm1.0}$ | $\underline{63.5_{\pm1.1}}$ | $\mathbf{66.1_{\pm1.7}}$ | $39.6_{\pm1.2}$ | $\underline{43.3_{\pm0.3}}$ | $\mathbf{43.5_{\pm0.6}}$ |
| | 50 | $78.2_{\pm0.7}$ | $\underline{82.3_{\pm1.1}}$ | $\mathbf{83.7_{\pm1.3}}$ | $59.8_{\pm0.8}$ | $\underline{65.0_{\pm0.8}}$ | $\mathbf{66.4_{\pm2.5}}$ |
| ResNet-18 | 10 | $64.9_{\pm0.6}$ | $\underline{66.2_{\pm1.2}}$ | $\mathbf{66.9_{\pm0.9}}$ | $42.2_{\pm1.2}$ | $\mathbf{47.2_{\pm1.6}}$ | $\underline{45.4_{\pm1.0}}$ |
| | 50 | $78.1_{\pm0.6}$ | $\underline{82.0_{\pm0.3}}$ | $\mathbf{82.5_{\pm0.7}}$ | $60.5_{\pm0.5}$ | $\underline{65.4_{\pm1.8}}$ | $\mathbf{65.8_{\pm1.3}}$ |

### A.3.2 GUIDANCE ON NOISY LATENT

We adopt the VP-SDE combined with the DDIM sampling algorithm, which enables a deterministic and efficient approximation of the reverse diffusion trajectory. A detail in this setting is the choice of the feature representation for prior guidance. Under DDIM dynamics, the conditional estimate of $\hat{z}_{0|t}$ can be expressed as an affine transformation of the current noisy state: $\hat{z}_{0|t} = \alpha_t z_t - \beta_t \epsilon_\theta(z_t, t)$, where $\alpha_t, \beta_t$ are deterministic coefficients and $\epsilon_\theta$ is the score predictor (Song et al., 2021). From the perspective of reverse dynamics, this relation holds as a first-order approximation under linearization, implying that the gradient fields induced by guiding $z_t$ and guiding $\hat{z}_{0|t}$ are approximately equivalent (see fig. 6). Hence, instead of explicitly computing the denoised estimation $\hat{z}_{0|t}$, we directly apply guidance on the noisy latent $z_t$, while avoiding the computational overhead of explicit decoding process at each timestep.

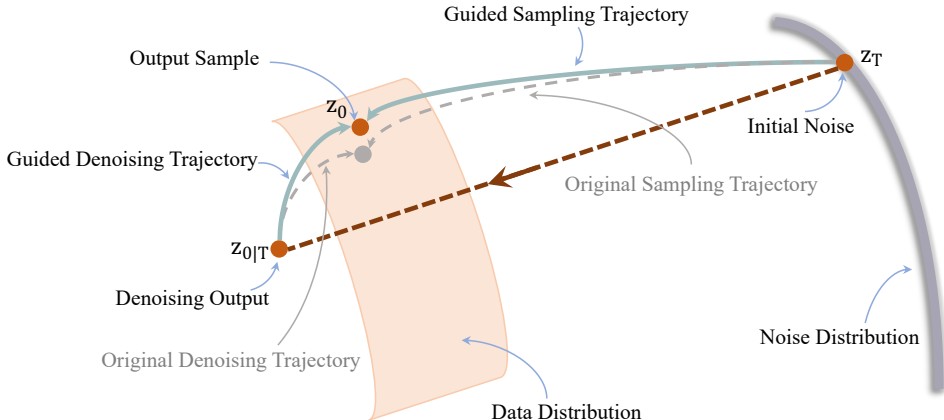

Figure 6: A sketch map of the relationship between $\hat{z}_{0|t}$ and $z_t$.

### A.3.3 KERNEL SELECTION

Besides the linear kernel, we also install our DAP with other Mercer kernels, such as the Radial Basis Function kernel (RBF, also known as the Gaussian kernel):

$$\mathcal{K}(x, y) = \exp\left(-\frac{\|x - y\|^2}{2\sigma^2}\right). \tag{21}$$

The bandwidth $\sigma$ controls the sensitivity: small $\sigma$ emphasizes fine-grained local features, whereas large $\sigma$ approaches the behavior of the linear kernel. Based on eq. (21), the induced distance becomes

$$\|\phi(x) - \phi(y)\|_{\mathcal{H}}^2 = \mathcal{K}(x, x) + \mathcal{K}(y, y) - 2\mathcal{K}(x, y) = 2 - 2\exp\left(-\frac{\|x - y\|^2}{2\sigma^2}\right), \tag{22}$$

which is a non-linear function of the Euclidean distance.

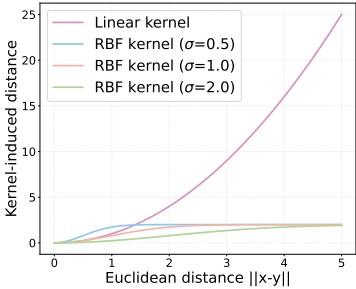

Figure 7: Curves of different Mercer kernel-induced distances.

Table 8: Top-1 Accuracy with different Mercer kernels. The results are evaluated on **ResNet-18** with **hard-label protocol**.

| Kernel | ImageNette | | ImageWoof | |
|---|---|---|---|---|
| | IPC10 | IPC50 | IPC10 | IPC50 |
| Linear | **66.4**$_{\pm0.5}$ | 82.8$_{\pm1.1}$ | 43.9$_{\pm0.9}$ | 63.2$_{\pm0.7}$ |
| RBF($\sigma = 0.5$) | 65.7$_{\pm0.2}$ | 82.5$_{\pm0.3}$ | **44.6**$_{\pm0.9}$ | **63.8**$_{\pm1.6}$ |
| RBF($\sigma = 1$) | 64.1$_{\pm1.7}$ | **83.1**$_{\pm0.8}$ | 44.1$_{\pm1.3}$ | 60.9$_{\pm0.5}$ |
| RBF($\sigma = 2$) | 66.0$_{\pm1.3}$ | 82.2$_{\pm1.0}$ | 43.7$_{\pm1.1}$ | 62.6$_{\pm2.2}$ |

As drawn in fig. 7, the RBF-induced distance grows quickly for small differences and saturates for large differences, effectively compressing large deviations while being sensitive to local differences. Table 8 reveals that the distillation performance of RBF kernel is comparable to that of linear kernel. To avoid introducing additional hyperparameters, we recommend using the linear kernel due to its simplicity and tractability.

### A.3.4 COMPARISON WITH RECENT METHODS

Table 9 presents the distillation results for ImageNet subsets on DiT. $CaO_2$ surpasses DAP in several cases, but this advantage arises from differences in evaluation protocols. The $CaO_2$ paper reports the best accuracy across soft-label and hard-label protocols, selecting the protocol that yields a higher performance. In contrast, $D^3HR$, VLCP, and DAP consistently adopt the hard-label protocol in this experiment. Since the soft-label protocol typically leads to higher accuracy, the results of $CaO_2$ likely benefit from this more permissive approach. Under a consistent hard-label evaluation setting, DAP remains competitive and performs better in most cases compared to these recent methods.

Table 9: Top-1 Accuracy on **ImageNette** and **ImageWoof**. †: The results are evaluated with **hard-label protocol** except for $CaO_2$.

| Dataset | Model | IPC | Random | DM | DiT | $CaO_2^\dagger$ | $D^3HR$ | VLCP | DAP | Full |
|---|---|---|---|---|---|---|---|---|---|---|
| Nette | ConvNet-6 | 10 | $46.0_{\pm0.5}$ | $49.8_{\pm1.1}$ | $56.2_{\pm1.3}$ | - | - | - | $\mathbf{64.8}_{\pm0.8}$ | |
| | | 50 | $71.8_{\pm1.2}$ | $70.3_{\pm0.8}$ | $74.1_{\pm0.6}$ | - | - | - | $\mathbf{82.2}_{\pm1.6}$ | $94.3_{\pm0.5}$ |
| | | 100 | $79.9_{\pm0.8}$ | $78.5_{\pm0.8}$ | $78.2_{\pm0.3}$ | - | - | - | $\mathbf{85.7}_{\pm1.3}$ | |
| | ResNetAP-10 | 10 | $54.2_{\pm1.2}$ | $60.2_{\pm0.7}$ | $62.8_{\pm0.8}$ | - | - | $64.8_{\pm3.6}$ | $\mathbf{67.8}_{\pm1.2}$ | |
| | | 50 | $77.3_{\pm1.0}$ | $76.7_{\pm1.0}$ | $76.9_{\pm0.5}$ | - | - | $81.2_{\pm0.8}$ | $\mathbf{82.3}_{\pm0.7}$ | $94.6_{\pm0.5}$ |
| | | 100 | $81.1_{\pm0.6}$ | $80.9_{\pm0.7}$ | $80.1_{\pm1.1}$ | - | - | - | $\mathbf{86.0}_{\pm2.1}$ | |
| | ResNet-18 | 10 | $55.8_{\pm1.0}$ | $60.9_{\pm0.7}$ | $62.5_{\pm0.9}$ | $65.0_{\pm0.7}$ | - | - | $\mathbf{66.4}_{\pm0.5}$ | |
| | | 50 | $75.8_{\pm1.1}$ | $75.0_{\pm1.0}$ | $75.2_{\pm0.9}$ | $84.5_{\pm0.6}$ | - | - | $\mathbf{82.8}_{\pm1.1}$ | $95.3_{\pm0.6}$ |
| | | 100 | $82.0_{\pm0.4}$ | $81.5_{\pm0.4}$ | $77.8_{\pm0.6}$ | - | - | - | $\mathbf{85.5}_{\pm1.5}$ | |
| Woof | ConvNet-6 | 10 | $25.2_{\pm1.1}$ | $27.6_{\pm1.2}$ | $32.3_{\pm0.8}$ | - | - | $34.8_{\pm2.4}$ | $\mathbf{37.6}_{\pm0.9}$ | |
| | | 50 | $41.9_{\pm1.4}$ | $43.8_{\pm1.1}$ | $48.5_{\pm1.3}$ | - | - | $54.5_{\pm0.6}$ | $\mathbf{55.8}_{\pm0.4}$ | $85.9_{\pm0.4}$ |
| | | 100 | $52.3_{\pm1.5}$ | $50.1_{\pm0.9}$ | $54.2_{\pm1.5}$ | - | - | $\mathbf{62.7}_{\pm1.4}$ | $62.4_{\pm1.2}$ | |
| | ResNetAP-10 | 10 | $31.6_{\pm0.8}$ | $29.8_{\pm1.0}$ | $39.0_{\pm0.9}$ | - | $40.7_{\pm1.0}$ | $39.5_{\pm1.5}$ | $\mathbf{41.8}_{\pm0.7}$ | |
| | | 50 | $50.1_{\pm1.6}$ | $47.8_{\pm1.2}$ | $55.8_{\pm1.1}$ | - | $59.3_{\pm0.4}$ | $57.3_{\pm0.5}$ | $\mathbf{63.3}_{\pm0.5}$ | $87.2_{\pm0.6}$ |
| | | 100 | $59.2_{\pm0.9}$ | $59.8_{\pm1.3}$ | $62.5_{\pm0.9}$ | - | $64.7_{\pm0.3}$ | $65.7_{\pm0.5}$ | $\mathbf{70.8}_{\pm1.4}$ | |
| | ResNet-18 | 10 | $30.9_{\pm1.3}$ | $30.2_{\pm0.6}$ | $40.6_{\pm0.6}$ | $45.6_{\pm1.4}$ | $39.6_{\pm1.0}$ | $39.9_{\pm2.6}$ | $\mathbf{43.9}_{\pm0.9}$ | |
| | | 50 | $54.0_{\pm0.8}$ | $53.9_{\pm0.7}$ | $57.4_{\pm0.7}$ | $68.9_{\pm1.1}$ | $57.6_{\pm0.4}$ | $58.9_{\pm1.5}$ | $\mathbf{63.2}_{\pm0.7}$ | $89.0_{\pm0.6}$ |
| | | 100 | $63.6_{\pm0.5}$ | $64.9_{\pm0.7}$ | $62.3_{\pm0.5}$ | - | $66.8_{\pm0.6}$ | $68.3_{\pm0.4}$ | $\mathbf{71.6}_{\pm1.3}$ | |

### A.3.5 CROSS-DATASET GENERALIZATION

We posit that the cross-datasets evaluation is essential to measure the generalization and versatility of a DD method, which is overlooked by most DD methods. According to Su et al. (2024), we extract 200 categories, which are predefined in Le & Yang (2015), from the ImageNet-1K dataset distilled by DAP as the distilled Tiny-ImageNet dataset. Table 10 shows that the extracted subsets (end with "-G") maintain strong validation performance on the target set, thereby confirming that our distilled data not only preserves the utility of the original dataset but also supports effective reuse across datasets. The results highlight the advantage of DAP: the distilled dataset is not tied to a single dataset domain but can be flexibly transferred and reused.

Table 10: Top-1 Accuracy on **Tiny ImageNet** (IPC50). The results are evaluated with **soft-label protocol**.

| Method | ResNet-18 | ResNet-50 | ResNet-101 |
|---|---|---|---|
| Full | 61.9 | 62.0 | 62.3 |
| $SRe^2L$ | 44.0 | 47.7 | 49.1 |
| $D^4M$ | 46.2 | 51.8 | 51.0 |
| $D^4M$-G | 46.8 | 51.9 | 53.2 |
| DAP-G | $\mathbf{50.3}_{\pm1.8}$ | $\mathbf{53.6}_{\pm1.0}$ | $\mathbf{54.7}_{\pm1.6}$ |

### A.3.6 EARLY STOP STRATEGY

In the guided sampling process, we employ the early stop guidance mechanism, which enhances sampling quality by only guiding earlier diffusion timesteps than the entire timesteps, thereby providing a better trade-off between sample diversity and fidelity (Chen et al., 2025; Santiago et al.,

2025). Besides, applying representativeness prior guidance in the early stage of the sampling trajectory also reduces the sampling cost (refer to section A.3.7). We summarize the sampling process with an early stop strategy in algorithm 2. To evaluate its effectiveness, we conducted experiments with different stopping parameters $t_{stop}$. The mechanism deactivates guidance for timesteps $t < t_{stop}$ in the reverse process, $t_{stop} = 0$ means complete guidance, while $t_{stop} = 50$ represents no guidance. The qualitative and quantitative results, as illustrated in fig. 8, indicate that $t_{stop} = 25$ yields the best performance.

---

**Algorithm 2** DAP Sampling with Early Stop(VP-SDE)

---

**Require:** Noisy data samples $\boldsymbol{x}_t^{train|c}$ within class $c$, pre-trained diffusion model $\boldsymbol{\epsilon}_\theta$, a layer output $\phi$ from diffusion backbone network, a Mercer Kernel induced distance measurement $d$, energy-based guidance scale $\gamma$, pre-defined noise scales $\beta_t$, early stop parameter $t_{stop}$.

1: $\boldsymbol{x}_T \sim \mathcal{N}(0, I)$
2: **for** $t = T, \cdots 1$ **do**
3:    $\boldsymbol{\epsilon} \sim \mathcal{N}(0, I)$ **if** $t > 1$, **else** $\boldsymbol{\epsilon} = \boldsymbol{0}$
4:    $\tilde{\boldsymbol{x}}_{t-1} = (2 - \sqrt{1 - \beta_t})\boldsymbol{x}_t + \beta_t \boldsymbol{\epsilon}_\theta(\boldsymbol{x}_t, t) + \sqrt{\beta_t}\boldsymbol{\epsilon}$
5:    **if** $t \leq t_{stop}$ **then**
6:        $\boldsymbol{x}_{t-1} = \tilde{\boldsymbol{x}}_{t-1}$                                            # Stop Guidance
7:    **else**
8:        $\boldsymbol{z}_t = \phi(\boldsymbol{x}_t)$, $\boldsymbol{z}_t^{train|c} = \phi(\boldsymbol{x}_t^{train|c})$          # Diffusion as representativeness priors
9:        $\boldsymbol{g}_t = -\nabla_{\boldsymbol{x}_t} d(\boldsymbol{z}_t, \boldsymbol{z}_t^{train|c})$
10:       $\boldsymbol{x}_{t-1} = \tilde{\boldsymbol{x}}_{t-1} + \gamma \boldsymbol{g}_t$                          # Guided sampling
11:   **end if**
12: **end for**
**Output:** $\boldsymbol{x}_0$                                            # The distilled sample of class $c$.

---

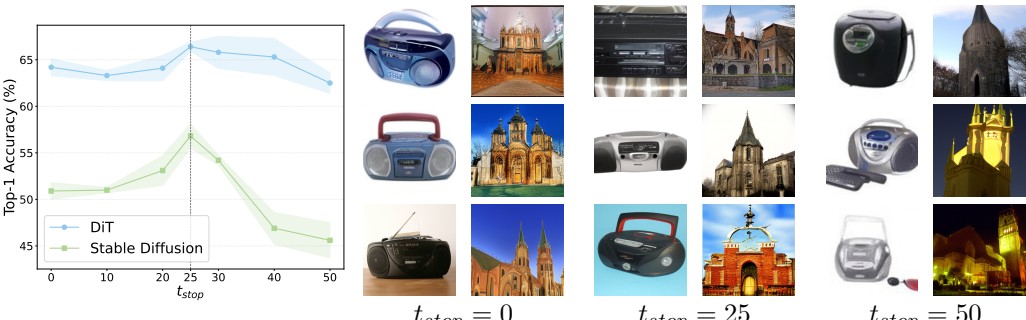

(a) Top-1 Accuracy on **ImageNette** under different $t_{stop}$. The results are evaluated on **ResNet-18** with **hard-label protocol**.

(b) Visualizations of the distilled samples with different $t_{stop}$.

Figure 8: Ablation study on $t_{stop}$ selection.

### A.3.7 SAMPLING-TIME SCALING

DAP does not introduce additional training costs, since no external pre-training or fine-tuning is required. The representativeness prior is directly derived from the pre-trained diffusion backbone. However, to inject this prior during sampling and improve data quality, we must extract features from the noisy training data $\boldsymbol{x}_t^{train}$ using the backbone network $\phi$. This step inevitably brings additional sampling time overhead, which must be acknowledged.

To quantify this overhead, we report the GPU memory and the sampling speed for different data sizes in table 11. While sampling-time scaling introduces overhead, the cost remains manageable and predictable on single GPU card.

Table 11: The overhead of sampling-time scaling ($t_{stop} = 25$). The Top-1 Accuracy is evaluated on **ImageNet-1K** with **hard-label protocol** (IPC10). The memory and speed are reported on $1\times$ A40.

| Data Size | Stable Diffusion | | | DiT | | |
|---|---|---|---|---|---|---|
| | GPU Mem.(GB) | Speed(s/iter) | Acc(%) | GPU Mem.(GB) | Speed(s/iter) | Acc(%) |
| 500 | | 35.9 | $32.1_{\pm0.5}$ | | 15.3 | $44.6_{\pm2.4}$ |
| 1000 | 23.1 | 39.9 | $39.9_{\pm1.3}$ | 10.6 | 24.0 | $48.8_{\pm1.7}$ |
| 1500 | | 47.1 | $40.7_{\pm1.5}$ | | 32.3 | $49.1_{\pm1.2}$ |

## A.4 VISUALIZATIONS

### A.4.1 REPRESENTATIVENESS GUIDANCE SCALE

As suggested by our ablation results (see figs. 5c and 5d in section 4.4), increasing $\gamma$ within a moderate scale effectively boosts representativeness prior, leading to improved downstream performance. However, excessive $\gamma$ introduces adverse effects. Over-amplifying the representativeness prior distorts the sampling trajectory, resulting in over-constrained generations that sacrifice diversity and generalization. Since the gradients of the other two priors are fixed, an imbalanced emphasis on representativeness suppresses their contribution, yielding biased and less informative images. This trade-off is clearly visualized in fig. 9.

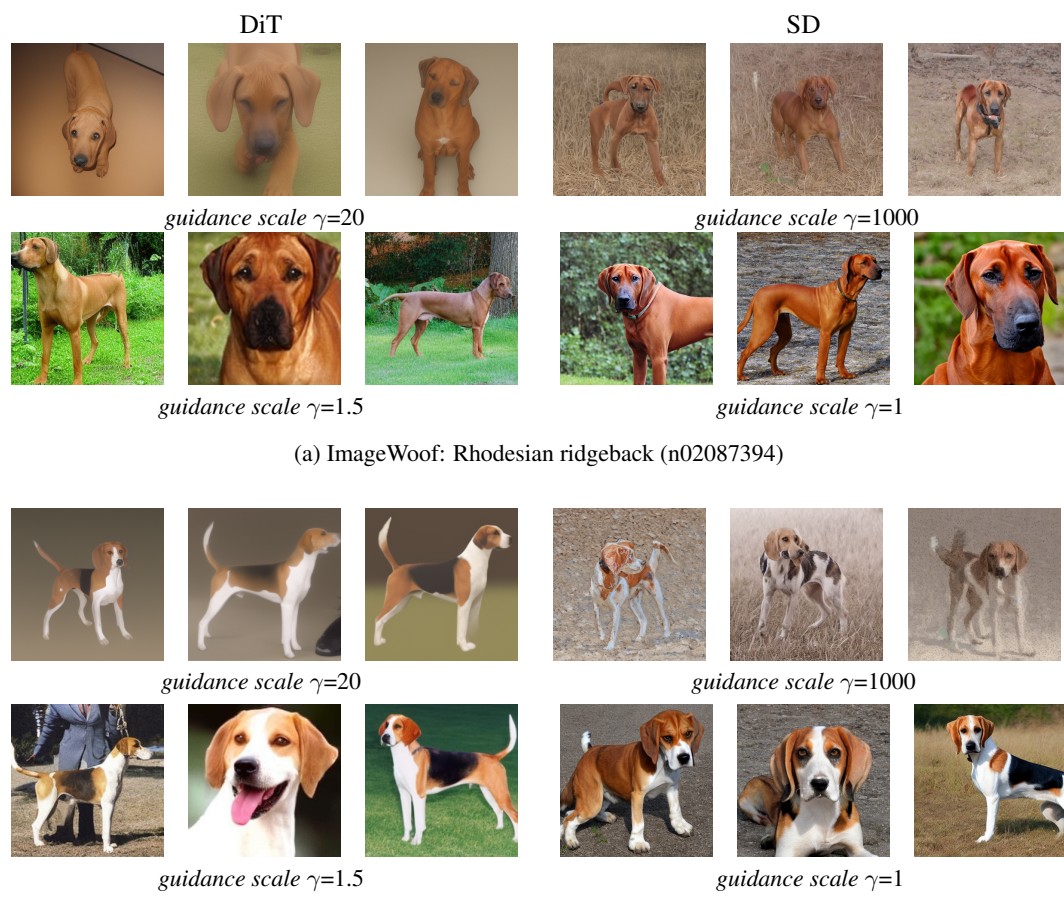

DiT  SD

*guidance scale $\gamma$=20*  *guidance scale $\gamma$=1000*

*guidance scale $\gamma$=1.5*  *guidance scale $\gamma$=1*

(a) ImageWoof: Rhodesian ridgeback (n02087394)

*guidance scale $\gamma$=20*  *guidance scale $\gamma$=1000*

*guidance scale $\gamma$=1.5*  *guidance scale $\gamma$=1*

(b) ImageWoof: English foxhound (n02089973)

Figure 9: Samples distilled by DiT (left three columns) and SD (right three columns). The excessive representativeness guidance scale $\gamma$ will generate representativeness bias in the sampling trajectory, affecting the diversity and fidelity of the synthetic images.

### A.4.2 REPRESENTATIVENESS COMPARISON

To provide an intuitive comparison, we visualize the distilled datasets obtained from different methods, as shown in fig. 10. For each group, we compute the distance measure defined in eq. (5) and report its representativeness ($\propto \frac{1}{d(\phi(x),\phi(y))}$). The results demonstrate that while all methods can preserve semantic information thanks to the powerful DMs, the images distilled by DAP consistently achieve the highest representativeness. This highlights the advantage of DAP in generating distilled datasets that are not only semantically valid but also representative.

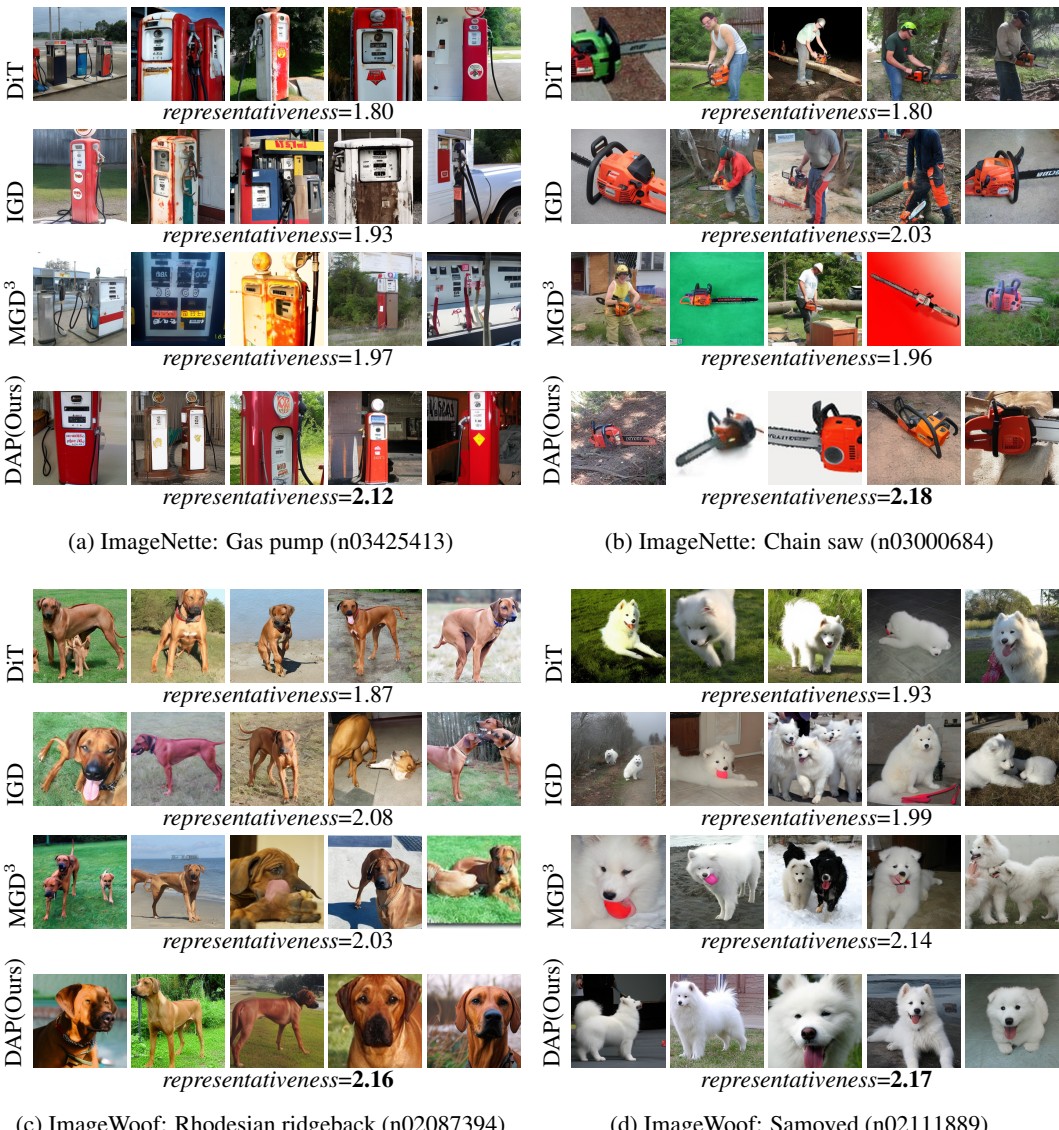

Figure 10: Visualization results of different DD methods. At the bottom of each group, we use the pre-trained DiT to calculate the average representativeness values ($\times 10^{-2}$).

### A.5 DISCLOSURE OF THE USE OF LARGE LANGUAGE MODELS

Given that the use of large language models (LLMs) is allowed as a general-purpose assist tool, this work utilizes LLMs to polish the sentences of the article. There is **no** significant role in research ideation and writing to the extent that they cannot be regarded as contributors.

