# OpenReview forum: "Diffusion Models as Dataset Distillation Priors"
_ICLR.cc/2026/Conference — ICLR 2026 Poster_

### Official Review · Reviewer_URQY · 2025-10-25

**Soundness:** 3
**Presentation:** 2
**Contribution:** 3
**Rating:** 6
**Confidence:** 4

**Summary:**

This paper interprets diffusion models as possessing three key characteristics: diversity, generalization, and representativeness. Building on this interpretation, the authors propose a training-free framework aimed at enhancing the representativeness of diffusion models. The improvement is achieved by incorporating a distance metric based on the Mercer kernel into the backward process of diffusion models.

**Strengths:**

1. The proposed DAP method substantially improves the quality of distilled datasets and leads to higher test accuracy.
2. The paper provides a thorough theoretical analysis that clearly establishes the connections between diffusion priors and the dataset distillation (DD) task..

**Weaknesses:**

1. Sections 3.1 and 3.2 mainly restate that diffusion models exhibit diversity and generalization, which are well-known properties. This content would be more appropriate as background material in the introduction rather than the method section, as it does not present any novel contributions.
2. The introduction of the kernel function in Section 3.3.1 lacks clarity. Its purpose and motivation are not well explained, and the connection between representativeness and the kernel function is missing. Although its application becomes evident in Section 3.3.2, the earlier introduction in 3.3.1 is confusing when read sequentially.
3. In Algorithm 1, the procedure describes how to generate a single sample using representative guidance. However, it is unclear how multiple synthetic images per class (i.e., IPC images) are generated. Additionally, the explanation of how these generated images collectively maintain the three claimed characteristics (diversity, generalization, and representativeness) remains insufficient, given the inherent randomness in individual generations.

**Questions:**

Why didn’t the authors provide an analysis of the computational cost? Since the proposed framework is claimed to be training-free, it should inherently offer computational advantages over training-based methods. However, without quantitative comparisons (e.g., runtime, memory usage, or efficiency metrics).

---

> ### Author Response · Authors · 2025-11-21
> **Response to Reviewer URQY [Part 1/2]**
>
> We sincerely thank the reviewer for your insightful comments. We have polished the paper, made the clarifications in the revised version. Below, we would like to provide point-to-point responses to all the raised questions:
>
> > **Q1: Sections 3.1 and 3.2 mainly restate that diffusion models exhibit diversity and generalization, which are well-known properties. This content would be more appropriate as background material in the introduction rather than the method section, as it does not present any novel contributions.**
>
> **A1**: We agree that the diversity and generalization properties of diffusion models are well-established in the literature. Since Section 3 is titled 'Diffusion As Priors' and Section 3.1 introduces the overall motivation of the method, the intention is to keep Section 3.2 concise while providing minimal context for the reader. An example (Table 1) using commonly used distillation datasets is included to validate these inherent priors. The main technical development begins in Section 3.3.
>
> It is noted that completely removing Section 3.2 and introducing representativeness immediately after the motivation would make the narrative abrupt and incomplete. To balance completeness and readability, the submission is revised as follows:
> - Section 3.2 is further condensed to retain only content directly tied to the motivation of DD and the explanation around Table 1,
> - A new subsection is added in Section 2 (Preliminary) to summarize the well-established diversity and generalization priors in diffusion models.
>
> These revisions are expected to strengthen the logical flow and improve overall clarity as recommended.
>
>
> > **Q2: The introduction of the kernel function in Section 3.3.1 lacks clarity. Its purpose and motivation are not well explained, and the connection between representativeness and the kernel function is missing. Although its application becomes evident in Section 3.3.2, the earlier introduction in 3.3.1 is confusing when read sequentially.**
>
> **A2**: We agree that the introduction of the kernel function in Section 3.3.1 appeared somewhat sudden and would benefit from a more gradual explanation. In the revision, this subsection is restructured to first introduce the kernel function and explain why kernel functions are beneficial for quantifying representativeness.
>
> Specifically, since representative samples refer to a subset that accurately reflects the characteristics of the larger population from which it is drawn, kernel functions provide a principled way to measure this reflection between synthetic and real data. For summary, the kernel function acts as a bridge between the representativeness prior and generated samples: it enables (a) similarity measurement through its induced distance, and (b) injection of this measure as a differentiable energy term into the sampling process.
>
> > **Q3: In Algorithm 1, the procedure describes how to generate a single sample using representative guidance. However, it is unclear how multiple synthetic images per class (i.e., IPC images) are generated.**
>
> **A3**: Theoretically, generating multiple IPC samples requires sampling multiple initial noises in the latent space and applying our DAP denoising process to each trajectory. In practice, modern diffusion frameworks such as `diffusers` handle this by setting the `num_images_per_prompt (int=IPC)` parameter in the pipeline, which automatically triggers batched and parallel generation to produce all IPC samples for a class in a single forward pass.

---

> ### Author Response · Authors · 2025-11-21
> **Response to Reviewer URQY [Part 2/2]**
>
> > **Q4: Additionally, the explanation of how these generated images collectively maintain the three claimed characteristics (diversity, generalization, and representativeness) remains insufficient, given the inherent randomness in individual generations.**
>
> **A4**: Regarding how the generated samples collectively maintain the three claimed properties, we clarify this more explicitly in the revised submission (Line286-293).
> Specifically, diversity naturally arises from the stochasticity of diffusion trajectories, where different noise initializations lead to distinct denoising paths, ensuring broad coverage of the data manifold without requiring explicit diversity regularization. Generalization is a dataset-level property that measures how well the synthetic dataset captures the structure of the real distribution, with this ability stemming from the pretrained diffusion score $\nabla_x \log p(x)$. Representativeness is directly enforced by our energy term in Eq. (8), which continually guides each trajectory toward feature regions that are well represented by real data.
>
> Empirically, the t-SNE visualizations (see Fig. 4) show that the full synthetic dataset exhibits smooth embedding structures and maintains class separability, indicating that diversity and generalization are preserved. Together, these mechanisms ensure that multiple IPC samples collectively exhibit diversity, generalization, and representativeness.
>
> > **Q5: Why didn’t the authors provide an analysis of the computational cost? Since the proposed framework is claimed to be training-free, it should inherently offer computational advantages over training-based methods. However, without quantitative comparisons (e.g., runtime, memory usage, or efficiency metrics).**
>
> **A5**: Runtime and GPU memory usage of DAP are reported in Table 10, but no direct comparison is made with other approaches, as such comparisons would be unfair for the following reason:
>
> If efficiency is evaluated only by sampling speed, DAP is indeed slower than the baselines because it compares each denoising trajectory with as many real samples as possible to align and enhance representativeness. Furthermore, the memory cost is constant with respect to dataset size and depends only on the diffusion architecture (U-Net or DiT).
>
> The key reason of the unfairness is that DAP is a sampling-time scaling method, relying solely on the intrinsic capabilities of generative model itself without introducing any external modules or algorithms. In contrast, many baselines incur substantial before-sampling overhead:
>
> - IGD [R1] requires a classifier pretrained on the target dataset (e.g., ResNet-18 on ImageNet-1K) to compute influence functions, which introduces non-negligible computation before sampling begins.
>
> - $D^4M$ [R2] and $MGD^3$ [R3] both require clustering (e.g., K-Means) to identify prototypes or modes before sampling, which also incurs significant cost.
>
> - Minimax [R4] must fine-tune and adapt the diffusion model using their proposed loss, leading to even higher distillation overhead.
>
> These costs are substantial and cannot be ignored, which is why reporting only sampling-time overhead comparisons would be misleading.
>
> ---
> **References**
>
> [R1]: Mingyang Chen et al. Influence-guided diffusion for dataset distillation. ICLR2025.
>
> [R2]: Duo Su et al. $D^4M$: Dataset distillation via disentangled diffusion model. CVPR2024.
>
> [R3]: Jeffrey A Chan-Santiago et al. $MGD^3$: Mode-guided dataset distillation using diffusion models. ICML2025.
>
> [R4]: Jianyang Gu et al. Efficient dataset distillation via minimax diffusion. CVPR2024.
>
> ---
> We believe these revisions fully address the questions raised. Please don’t hesitate to let us know if there are any additional clarifications or experiments that we can offer!

---

> ### Comment · Reviewer_URQY · 2025-11-27
>
> Thank you for the detailed rebuttal and clarifications. The responses have addressed my concern, and I will maintain my current positive score.

---

> > ### Author Response · Authors · 2025-11-28
> > **Thanks for the feedback**
> >
> > Thank you again for your valuable comments and feedback, which improved our paper. We highly appreciate it.

---

### Official Review · Reviewer_1VVS · 2025-10-29

**Soundness:** 2
**Presentation:** 2
**Contribution:** 3
**Rating:** 6
**Confidence:** 2

**Summary:**

The paper introduces a novel framework that employs diffusion models as powerful generative priors for dataset distillation, the task of compressing a large dataset into a compact synthetic subset while preserving downstream task performance. Unlike traditional approaches that directly optimize synthetic samples in pixel or feature space, this method leverages the diffusion process to model the underlying data manifold, enabling the generation of representative and diverse samples without costly bi-level optimization. Furthermore, the framework enhances representativeness by encouraging the synthetic data to align with real samples in the feature space via a Mercer kernel-based similarity measure. Experimental results on the ImageNet benchmark demonstrate that the proposed approach consistently outperforms state-of-the-art baselines in classification accuracy.

**Strengths:**

The paper presents a comprehensive literature review and demonstrates promising experimental results, highlighting the effectiveness and practical potential of the proposed approach.

**Weaknesses:**

Although the proposed method is straightforward and effective, several key implementation details are missing, making the reported performance difficult to reproduce. For instance, the paper does not clearly explain how the training samples are selected to pair with synthetic samples when computing the kernel distance.

**Questions:**

1. **Clarification of $x^{\text{train|c}}_t$ selection.**
In Algorithm 1, how are the samples $x^{\text{train|c}}_t$ obtained or selected before the DAP sampling process?


2. **Diversity Comparison with Baselines.**
The proposed method (Diffusion as Prior, DAP) demonstrates substantially greater diversity compared to baseline methods such as MGD³ and IGD, as illustrated in Figure 1. Interestingly, while MGD³ explicitly incorporates mechanisms to enhance diversity, the proposed method does not directly encourage the generation of diverse samples. Could the authors elaborate on the underlying reasons for this observed improvement in diversity?

3. **Quantitative and Qualitative Comparison.**
Could the authors provide a more detailed comparison between DAP and the baseline methods in terms of diversity, representativeness, and classification performance under different IPC (images per class) settings?

---

> ### Author Response · Authors · 2025-11-21
> **Response to Reviewer 1VVS**
>
> We sincerely thank the reviewer for your insightful comments. We have polished the paper, added the experiment results and made the clarifications in the revised version. Below, we would like to provide point-to-point responses to all the raised questions:
>
> > **Q1: Clarification of $x_t^{train|c}$ selection.**
>
> **A1**: DAP does not require selecting or pairing specific $x^{train|c}$ samples before the sampling process. For each class, all real training images are passed through the VAE encoder to obtain latent embeddings, which will be provided for representativeness guidance.
> At each denoising step, DAP computes the representativeness energy using Eq. (7) by comparing the current latent $x_t$ with all training samples from class $c$.
> The representativeness are averaged and transformed directly into the conditional score function for the current sampling step. Thus, representativeness guidance is applied continuously along the denoising trajectory, without any pairing heuristic or sample selection strategy. For GPU efficiency, real samples within class $c$ are split into mini-batches, and features are extracted batch by batch. The implementation description will be revised to make this process explicit.
>
>
> > **Q2: Diversity Comparison with Baselines.**
>
> **A2**: $MGD^3$ and $D^4M$ rely on K-means clustering, which is sensitive to initialization. Besides, the K-means clustering are based on Euclidean distance, thus, the learned centroids face the risk of falling out of the data manifold, which is unable to reflect the underlying data manifold and lead to semancitc inconsistency [R1].
> As a result, prototypes for generation guidance may inherit this geometric limitation. Furthermore, the inductive bias in K-means emphasizes representative centroids over diversity, often leading to mode-limited prototypes [R2].
>
> In contrast, DAP uses a mercer kernel-induced distance in the feature space. This metric provides a more meaningful similarity measure on the data manifold, enabling guidance toward representative regions without constraining trajectories to a finite set of centroids (i.e., prototypes).
>
> Diversity and representativeness require a trade-off during sampling. DAP explicitly controls this trade-off through the parameter $\gamma$, which adjusts the strength of representativeness guidance. K-means–based methods implicitly incorporate the trade-off in the clustering step. Although $MGD^3$ includes a trade-off parameter, it directly controls mode guidance rather than representativeness or diversity guidance. These points will be clarified in the revision.
>
> ---
> **References**
>
> [R1]: Ergun J C et al. Learning-Augmented $ k $-means Clustering. ICLR2022.
>
> [R2]: Jain A K et al. Data clustering: a review. ACM computing surveys, 1999, 31(3): 264-323.
>
>
> > **Q3: Quantitative and Qualitative Comparison.**
>
> **A3**: The classification results under various IPC settings are presented in the main experimental sections. Here we focus on clarifying the quantitative findings on diversity and representativeness priors.
>
> Table1: Diversity comparison with FID ($\downarrow$) score.
>
> | Dataset | IPC | DiT | IGD | MGD^3 | DAP |
> | :----| :---: |:---: | :---: | :---: | :---: |
> | Nette | 10 | 106.9 | 106.8 | 106.1 | 105.6 |
> |  | 50 | 83.4 | 72.3 | 72.9 | 68.1|
> |  | 100 | 81.5 | 67.2 | 66.4 | 64.1 |
> | Woof | 10 | 270.6 | 265.4 | 262.4 | 261.7 |
> |  | 50 | 247.2 | 243.1 | 237.4 | 237.6 |
> |  | 100 | 242.1 | 238.9 | 231.6 | 225.2 |
>
> Table2: Representativeness comparison with Mercer kernel-induced distance ($\propto\frac{1}{d} \times10^{-2}$, $\uparrow$).
> | Dataset | IPC | DiT | IGD | MGD^3 | DAP |
> | :----| :---: |:---: | :---: | :---: | :---: |
> | Nette | 10 | 1.68 | 1.77 | 1.90 | 2.08|
> |  | 50 | 1.53 | 1.71 | 1.84 | 1.96|
> |  | 100 | 1.49 | 1.68 | 1.72 | 1.88|
> | Woof | 10 | 2.47 | 2.55 | 2.49 | 2.71|
> |  | 50 | 2.21 | 2.38 | 2.29 | 2.65|
> |  | 100 | 2.18 | 2.33 | 2.26 | 2.47|
>
> For diversity, Table1 presents FID scores across IPC settings. DAP consistently outperforms all baselines, with FID scores monotonically decreasing as IPC increases. This indicates that the distilled dataset becomes more diverse with additional synthesized images.
>
> For representativeness, Table2 displays the kernel-induced distance metric used in DAP. Since DAP directly optimizes this metric during sampling, it achieves the best representativeness across all IPC settings.
>
> An interesting trend is that all methods exhibit slightly worse average representativeness as IPC increases. We attribute this observation to synthesizing more samples causes the dataset to cover more modes, making each sample less representative of the full class distribution. This observation supports the discussion in Q2 on the need to balance diversity and representativeness. The abone analysis is clarified in the revision.
>
> ---
> We hope these revisions fully address the questions raised. Please don’t hesitate to let us know if there are any additional clarifications or experiments that we can offer!

---

> ### Comment · Reviewer_1VVS · 2025-11-25
>
> The author has addressed all my concerns, and I remain positive about this work.

---

> > ### Author Response · Authors · 2025-11-27
> > **Thanks for the feedback**
> >
> > Thank you again for your valuable comments and feedback, which improved our paper. We highly appreciate it.

---

### Official Review · Reviewer_wTWR · 2025-10-29

**Soundness:** 4
**Presentation:** 3
**Contribution:** 3
**Rating:** 4
**Confidence:** 3

**Summary:**

This paper proposes Diffusion As Priors (DAP), a novel framework for dataset distillation that leverages inherent priors in pre-trained diffusion models. The authors establish a theoretical connection between diffusion model objectives and dataset distillation requirements, identifying three key priors: diversity, generalization, and representativeness. The main contribution is formalizing representativeness using Mercer kernel-induced distances and incorporating this as guidance during the reverse diffusion process without requiring model retraining. Extensive experiments on ImageNet-1K and its subsets demonstrate that DAP achieves state-of-the-art performance while maintaining cross-architecture generalization.

The key novelty is decomposing the conditional score function as: $\nabla_x \log p(x|R) = \nabla_x \log p(x) + \nabla_x \log p(R|x)$, where the first term captures diversity/generalization priors from the pre-trained diffusion model, and the second term introduces representativeness through energy-based guidance using kernel-induced distance measurements.

**Strengths:**

- The paper provides a principled framework connecting diffusion model objectives to dataset distillation requirements through clear mathematical formulations and proofs.

- Extensive experiments across multiple datasets (ImageNet-1K, ImageNette, ImageWoof, ImageIDC), architectures (ConvNet, ResNet, MobileNet, EfficientNet, Swin), and protocols (hard-label, soft-label) demonstrate broad applicability.

 - Unlike methods requiring fine-tuning or external training, DAP leverages pre-trained diffusion models directly, making it practical and computationally efficient (no additional training cost).

**Weaknesses:**

- While the Mercer kernel framework is mathematically sound, the paper does not convincingly argue why minimizing kernel-induced distance in feature space is the optimal objective for representativeness in DD. Alternative formulations (e.g., maximum mean discrepancy, optimal transport) could be equally valid.

- Table 10 reveals significant computational costs during sampling, with speed increasing from 15-36 seconds per iteration depending on data size. This overhead could be prohibitive for large-scale applications. The paper acknowledges this but doesn't propose solutions.

- The method requires access to the full training dataset during sampling for representativeness guidance, which somewhat limits the practical benefits of distillation

- The paper primarily uses linear kernels with brief exploration of RBF (Table 8). Other kernel choices  and their theoretical implications are not discussed.

**Questions:**

- Could you provide more justification for why kernel-induced distance is the right metric for representativeness? Have you considered alternative metrics like Maximum Mean Discrepancy (MMD) or Optimal Transport distances? How would these compare theoretically and empirically?

- Are there scenarios or dataset characteristics where DAP underperforms? For instance, does it struggle with fine-grained classification, imbalanced data, or out-of-distribution classes?

- The linear kernel is chosen "due to its tractability" but Table 8 shows RBF performs comparably. Could you discuss the theoretical implications of different kernel choices more deeply? Does kernel selection depend on dataset characteristics?

- When the method does not have access to full training samples during sampling but instead a handful of subset of them, how will the proposed method perform?

---

> ### Author Response · Authors · 2025-11-21
> **Response to Reviewer wTWR [Part 1/2]**
>
> We appreciate the reviewer’s questions, which have helped us improve the paper quality significantly! We have polished the paper and made the clarifications in the revised version. Below we would like to provide point-to-point responses to all the raised questions:
>
> > **Q1: Could you provide more justification for why kernel-induced distance is the right metric for representativeness? Have you considered alternative metrics like Maximum Mean Discrepancy (MMD) or Optimal Transport (OT) distances? How would these compare theoretically and empirically?**
>
> **A1**:
> - DAP v.s. MMD:
>
> A key conceptual distinction is that MMD measures distribution-level discrepancy between the entire real distribution $P$ and the synthetic distribution $Q$. Formally,
> $$MMD^2(P,Q)=\|\mu_P-\mu_Q\|^2.$$
> Computing MMD requires access to the full set of synthetic samples. However, during DD, samples are generated sequentially during the sampling process, and the distribution $Q$ is not available in advance, making it impossible to estimate $\mu_Q$. Therefore, MMD cannot function as a per-step guidance term in diffusion sampling.
>
> In contrast, DAP employs a point-to-distribution measure computed online at each denoising step using only real data embeddings. This approach aligns with step-wise diffusion dynamics, enabling sample-level guidance, whereas MMD is inherently unsuitable for such tasks.
>
>
> - DAP v.s. OT:
>
> OT also defines a distribution-level discrepancy:
> $$W_p(P,Q)=(\inf_{\gamma\in\Gamma(P,Q)}\mathbb{E}_{(x,y)\sim\gamma}\|x-y\|^p)^\frac{1}{p},$$
> which requires access to the full synthetic distribution $Q$ to compute the optimal transport plan $\gamma^*$. Moreover, the exact OT via linear programming and entropically regularized OT using the Sinkhorn algorithm involve $\mathcal{O}(n^2)$ to $\mathcal{O}(n^3)$ memory and computational costs, which rely on iterative solvers, and often exhibit numerical instability during backpropagation through transport plans. These properties make OT impractical for large-scale DD or step-wise diffusion guidance.
>
> - Why Mercer Kernel？
>
> The use of the Mercer kernel in DAP offers several practical advantages. Mercer kernels enable point-to-point similarity to be computed implicitly in a Reproducing Kernel Hilbert Space, where distances are geometrically meaningful and align with the semantic feature space of diffusion models. This yields a smooth, differentiable objective with efficient gradients, ideal for step-wise guidance in diffusion sampling. Additionally, kernel computations are computationally efficient and avoid the intensive optimization or transport-plan estimation required by distribution-level metrics like MMD or OT.
>
> Furthermore, MMD and OT provide valuable perspectives for dataset distillation. Exploring the integration of diffusion priors with full distribution-level metrics represents a promising research direction, and we plan to investigate these combinations in our future work.
>
> > **Q2: Table 10 reveals significant computational costs during sampling, with speed increasing from 15-36 seconds per iteration depending on data size. This overhead could be prohibitive for large-scale applications. The paper acknowledges this but doesn't propose solutions.**
>
> **A2**: We agree with the reviewer that DAP does not provide a speed advantage in sampling time. DAP is an inference-time scaling method that relies on the generative model's inherent capacity rather than training auxiliary generators or classifiers, making a slower sampling speed acceptable in this context.
>
> For large-scale dataset distillation applications, several acceleration techniques can significantly reduce runtime and are fully compatible with the DAP framework:
> - DDIM sampling [R1] for deterministic fast trajectories;
> - Euler A [R2], and Heun [R2] samplers for improved efficiency;
> - Fast ODE solvers such as DPM-Solver [R3] or few-step distillation methods like LCM [R4], which greatly shorten the sampling chain.
>
> Additionally, speed can be further enhanced using advanced GPUs, such as NVIDIA RTX 4090 or A100. The reported results were obtained on the A40, which is noticeably slower than these newer architectures.
>
> ---
> **Refences**
>
> [R1]: J Song et al. Denoising Diffusion Implicit Models. ICLR2021.
>
> [R2]: Tero Karras et al. Elucidating the design space of diffusion-based generative models. NeurIPS2022.
>
> [R3]: Cheng Lu et al. DPM-Solver: A Fast ODE Solver for Diffusion Probabilistic Model Sampling in Around 10 Steps. NeurIPS2022.
>
> [R4]: S Luo et al. Latent consistency models: Synthesizing high-resolution images with few-step inference. arxiv2023.

---

> ### Author Response · Authors · 2025-11-21
> **Response to Reviewer wTWR [Part 2/2]**
>
> > **Q3: The method requires access to the full training dataset during sampling for representativeness guidance, which somewhat limits the practical benefits of distillation.**
>
> **A3**: Results for DAP on ImageNet-1K under different accessible data sizes are reported in Table 10. As the available data decreases, distillation performance degrades because the representativeness guidance is computed from fewer real samples.
> The dataset distillation setting inherently assumes full access to the target data. Reducing the real dataset size falls under a different problem, such as distillation under partial observability. Future work will focus on this new task.
>
> > **Q4: The linear kernel is chosen "due to its tractability" but Table 8 shows RBF performs comparably. Could you discuss the theoretical implications of different kernel choices more deeply? Does kernel selection depend on dataset characteristics?**
>
> **A4**: Appendix A.4.3 provides a detailed discussion of several commonly used kernels and their empirical performance in DAP. In the main experiments, the linear kernel is adopted due to its simplicity, parameter-free nature, and effectiveness. The distillation performance of the linear kernel is on par with or surpasses that of other kernels without introducing additional hyperparameters.
>
> While more complex kernels such as RBF or polynomial kernels can be used within DAP, they require manual tuning of extra hyperparameters, such as bandwidth, bias, or degree. This complicates the pipeline and introduces sensitivity unrelated to the core mechanism.
>
> Importantly, kernel selection is independent of dataset characteristics, as features are determined primarily by the pre-trained diffusion backbone rather than raw data geometry. The paper has been updated to clarify these considerations.
>
>
> > **Q5: Are there scenarios or dataset characteristics where DAP underperforms? For instance, does it struggle with fine-grained classification, imbalanced data, or out-of-distribution classes?**
>
> **A5**:
> - Fine-grained classification.
>
> DAP performs well on fine-grained classification tasks, as demonstrated on Imagewoof, a challenging fine-grained classification subset of ImageNet containing ten highly similar dog breeds. The classification accuracy results indicate that DAP remains effective even when intra-class variability is small.
>
> - Imbalanced datasets.
>
> DAP is not affected by class imbalance because (a) distillation is performed at the class level, using $x_t^{train|c}$ for each category independently, and (b) the distilled dataset is perfectly balanced by the setting of IPC (image per class) setting, which mitigates the imbalance issue present in the original dataset.
>
> - Out-of-distribution classes.
>
> For out-of-distribution (OOD) classes, DAP may exhibit performance degradation. If the target classes are unseen by the pretrained diffusion model, the original score $\nabla_x\log p(x)$ in Eq. (4) provides limited priors, and only the representativeness term $\nabla_x\log p(R|x)$ carries information from in-distribution classes, as the backbones can extract meaningful features even for OOD class samples [R1, R2]. In such cases, the diffusion priors may not fully capture the diversity or generalization properties needed for high-quality distillation.
>
> ---
> **Reference**
>
> [R1]: Huanran Chen et al. Diffusion models are certifiably robust classifiers. NeurIPS2024.
>
> [R2]: Xingyi Yang et al. Diffusion model as representation learner. ICCV2023.
>
> ---
> We hope these revisions fully address the questions raised. Please don’t hesitate to let us know if there are any additional clarifications or experiments that we can offer!

---

> ### Author Response · Authors · 2025-11-27
> **Looking forward to further feedback**
>
> Dear Reviewer wTWR,
>
> Thank you again for your great efforts and comments. We have carefully addressed your concerns in detail. We believe the additional experiments, analysis, and explanation have significantly improved the quality and clarity of our submission. We hope you might find the response satisfactory and regard this as a sufficient reason to raise the score. As the discussion phase is about to close, we look forward to hearing from you about any further feedback. We will be delighted to clarify further concerns (if any).
>
> Best,
> Authors.

---

### Official Review · Reviewer_2soE · 2025-10-30

**Soundness:** 3
**Presentation:** 3
**Contribution:** 3
**Rating:** 6
**Confidence:** 3

**Summary:**

Inspired by the diffusion classifiers, it posits that the feature extraction capability inherent in a well-trained diffusion model itself constitutes a representativeness prior highly relevant to DD. It hypothesizes that high representativeness corresponds to high similarity between synthetic and original data in the representation space. To formalize this, it employs the Mercer kernel, a specific type of kernel function, to quantify the similarity within feature spaces. The Mercer kernel provides mathematical guarantees of convexity and tractability in optimization, ensuring that the representativeness prior is computationally feasible. Empirically, it defines the representativeness score function as an energy function based on Mercer kernel, which allows to inject the unused representativeness prior into the distilled data through guided sampling.

**Strengths:**

It proposes Diffusion As Priors (DAP) and applies it to datasets of varying scales, including large-scale ImageNet-1K and its small subsets.
Both quantitative and qualitative results show that DAP significantly enhances the quality of distilled datasets. It validates the theoretical connections between diffusion priors and DD task, while achieving competitive performance compared to other methods.

It prove the priors in the well-trained DMs meet the diversity and generalization requirements of DD. It derives the overlooked representativeness prior from DMs and formalize it into a kernel-induced distance, which guides the sampling dynamic and improves the quality of distilled datasets. It further shows that by introducing the desired priors, the distilled datasets have the same generalization and transferability as the original ones.

To investigate whether DAP enforces diversity and representativeness priors in the distilled datasets, it visualizes the data distribution using t-SNE alongside both the training and test sets. It reveals that the synthetic data aligns well with the training set while generalizing to the test set, demonstrating that the DAP can accurately match the underlying data manifold. Moreover, the embeddings show intra-class diversity and inter-class separability, indicating that the distilled datasets capture meaningful variability without sacrificing discriminability.

It conducts ablation studies to investigate the influence of feature layer selection in representativeness guidance. The cases consistently reveal that the final output layers are suboptimal for representativeness guidance, as they prioritize distribution alignment over representativeness.

**Weaknesses:**

Diffusion models are widely adopted in dataset distillation to extract features and obtain information. It is similar to use diffusion as priors in this paper. It is better to discuss related works and highlight the differences.

It is better to compare with more baslines such as [R1,R2,R3], which also adopts diffusion for dataset distillation.


[R1] CaO2: Rectifying Inconsistencies in Diffusion-Based Dataset Distillation

[R2] Taming Diffusion for Dataset Distillation with High Representativeness

[R3] Dataset Distillation via Vision-Language Category Prototype

**Questions:**

see the weakness.

---

> ### Author Response · Authors · 2025-11-21
> **Response to Reviewer 2soE**
>
> We sincerely appreciate the insightful review and are grateful for the recognition of the technical soundness of the proposed DAP method. Based on the suggestions provided, we carefully compared DAP with the latest related works. Below, we would like to respond to the raised questions:
> > **Q1: It is better to compare with more baslines such as [R1,R2,R3], which also adopts diffusion for dataset distillation.**
>
> **A1**: First, these papers are incorporated into the background and compatibility discussion in the revised submission. We highlights the difference and compatibility between DAP and these methods:
>
> - DAP v.s. [R1] ($CaO_2$):
> $CaO_2$ is a two-stage DD framework designed to alleviate inconsistencies in objectives and conditions.
> This method does not introduce an explicit representativeness prior. Instead, it identifies correct and high-confidence samples through a fixed sample selector, which can be regarded as an indirect external representativeness prior. Compared to DAP, $CaO_2$ can be described as a post-processing method, with Phase I and II occurring after image generation. Thus, it is compatible with DAP.
>
> - DAP vs. [R2]: R2 utilizes DDIM inversion to map latents to the Gaussian domain and aligns representative latents with a high-normality Gaussian distribution through a proposed sampling scheme. R2 estimates per-class Gaussian statistics and selects the best subset via a designed metric. DAP sampling is compatible with R2 during DDIM sampling.
>
> - DAP v.s. [R3]: R3 constructs text prototypes to enrich labels with semantic information and fine-tunes diffusion models with image-text pairs. Since DAP is a training-free method, DAP can be implemented on R3 fine-tuned diffusion model checkpoints, similar to Minimax.
>
> Finally, we also list the quantitative comparison results below:
>
> Table1: Top-1 Accuracy on ImageNet-1K.
> | Eval. Model | IPC | R1 | R2 | R3 | DAP |
> | :-----| :----: | :----: | :----: | :----: | :----: |
> | ResNet-18 | 10 | $46.1 _{\pm 0.2}$ | $44.3 _{\pm 0.3}$ | $46.7 _{\pm 0.4}$ | $49.1 _{\pm 1.2}$ |
> | ResNet-18 | 50 | $60.0 _{\pm 0.0}$ | $59.4 _{\pm 0.1}$ | $60.5 _{\pm 0.2}$ | $62.7 _{\pm 1.5}$ |
>
> Table2: Top-1 Accuracy on ImageNette.
> | Method | ResNetAP-10 | ResNetAP-10 | ResNet-18 | ResNet-18 |
> | :-----| :----: | :----: | :----: | :----: |
> |       | IPC10 | IPC50 | IPC10 | IPC50 |
> | R1 | - | - | $65.0_{\pm0.7}$ | $84.5 _{\pm 0.6}$ |
> | R3 | $64.8 _{\pm 3.6}$ | $81.2 _{\pm 0.8}$ | - | - |
> | DAP | $67.8 _{\pm 1.2}$ | $82.3 _{\pm 0.7}$ | $66.4 _{\pm 0.5}$ | $82.8 _{\pm 1.1}$ |
>
> Table3: Top-1 Accuracy on ImageWoof.
> | Eval. Model | IPC | R1 | R2 | R3 | DAP |
> | :-----| :----: | :----: | :----: | :----: | :----: |
> | ConvNet-6 | 10 | - | - | $34.8 _{\pm 2.4}$ | $37.6 _{\pm 0.9}$ |
> | ConvNet-6 | 50 | - | - | $54.5 _{\pm 0.6}$ | $55.8 _{\pm 0.4}$ |
> | ConvNet-6 | 100 | - | - | $62.7 _{\pm 1.4}$ | $62.4 _{\pm 1.2}$ |
> | ResNetAP-10 | 10 | - | $40.7 _{\pm 1.0}$ | $39.5 _{\pm 1.5}$ | $41.8 _{\pm 0.7}$ |
> | ResNetAP-10 | 50 | - | $59.3 _{\pm 0.4}$ | $57.3 _{\pm 0.5}$ | $63.3 _{\pm 0.5}$ |
> | ResNetAP-10 | 100 | - | $64.7 _{\pm 0.3}$ | $65.7 _{\pm 0.5}$ | $70.8 _{\pm 1.4}$ |
> | ResNet-18 | 10 | $45.6 _{\pm 1.4}$ | $39.6 _{\pm 1.0}$ | $39.9 _{\pm 2.6}$ | $43.9 _{\pm 0.9}$ |
> | ResNet-18 | 50 | $68.9 _{\pm 1.1}$ | $57.6 _{\pm 0.4}$ | $58.9 _{\pm 1.5}$ | $63.2 _{\pm 0.7}$ |
> | ResNet-18 | 100 | - | $66.8 _{\pm 0.6}$ | $68.3 _{\pm 0.4}$ | $71.6 _{\pm 1.3}$ |
>
> In Table1, all methods are evaluated under the soft-label protocol on ImageNet-1K, and DAP achieves the best result consistently.
> Table2 and Table3 list the subset distillation results. R1 surpasses DAP in some cases because R1 reports the best results regarless of the evaluation protocol. According to the R1 paper, the authors evaluate both soft-label and hard-label protocols and report the higher accuracy. Thus, the results of R1 may come from the soft-label evaluation, which generally performs better than hard-label protocol (i.e., R2, R3, and DAP report hard-label protocol results, while the protocol for R1 is unclear).
> Based on comparisons with these recent papers, DAP also performance better.
>
> ---
> We believe these revisions fully address the questions raised. Please don’t hesitate to let us know if there are any additional clarifications or experiments that we can offer!

---

### Author Response · Authors · 2025-11-21
**General Response**

We sincerely thank all reviewers for their constructive feedback, especially for recognizing the strengths and contributions of our work.
We have polished the paper, added the experiment results, and made clearer clarifications in the revised version. The updated version now includes:

- We have included clearer explanations of diffusion priors and improved presentation of Section 3.

- We have expanded intuition and motivation for the kernel-based representativeness prior.

- We have discussed how DAP can achieve the three properties collectively.

- We have included clarifications on sampling multiple IPC samples, feature computation, and implementation details.

- We have compared and analyzed DAP with the recent diffusion-based DD methods.

- We have improved our writing, figures, and organization.

All modifications in the main text and appendix are highlighted in **blue** for ease of reference.

We greatly appreciate the reviewers’ insights, which helped us significantly improve the clarity and impact of this work. Please let us know if any further clarification is needed.

---

### Author Response · Authors · 2025-12-02
**Global Response**

Dear Area Chair and Reviewers,

We would like to express our deepest gratitude for your diligent coordination in the review process and for the insightful feedback provided by the reviewers. During the author-reviewer discussion phase, we endeavored to address the concerns raised. As a few reviewers did not have time to engage in the discussion phase, we would like to summarize the concerns and responses regarding our work.

| **Dimension** | **Key Concern**  | **Our Response** |
| :--- | :--- | :--- |
|Clarification| Why Mercer kernels and why linear kernel ```(wTWR)```| The core distinction is that Mercer kernels enable **point-to-point** similarity to be computed implicitly in a Reproducing Kernel Hilbert Space, where distances are geometrically meaningful and align with the semantic feature space of diffusion models. In the main experiments, the linear kernel is adopted due to its **simplicity, parameter-free nature, and effectiveness**. |
|Clarification| Scenarios where DAP underperforms? ```(wTWR)``` | According to the experimental results, DAP **performs well** in fine-grained classification and imbalanced datasets. Even in OOD tasks, the proposed representativeness score estimation **helps the original generation pipeline**.|
|Clarification| Access to the full training dataset ```(wTWR)``` | The settings of DD inherently assume full access to the target dataset. Reducing the real dataset size falls under a different task. |
|Clarification| Diversity Comparison with Baselines ```(1VVS)``` | We argue that diversity and representativeness require a **trade-off** during sampling. DAP explicitly controls this trade-off through the parameter $\gamma$, which adjusts the strength of representativeness guidance.|
|Experimental Details| Comparison with recent SOTA methods ```(2soE)``` | DAP **performs better** compared with these recent works. Also, they are **compatible** with the proposed DAP.|
|Experimental Details| Efficiency & Cost ```(wTWR, URQY)``` | The added generation latency is modest and justified for **sampling(inference)-time scaling** since we have to work up the capability of the diffusion model itself rather than using other external models.|
|Experimental Details| Sample selection ```(1VVS)``` and generation ```(URQY)``` | DAP **does not** require selecting or pairing specific training samples, and generating multiple IPC samples requires **sampling multiple initial noises** in the latent space and applying our DAP denoising process to each trajectory. |
|Experimental Details| More quantitative and qualitative comparison ```(1VVS)``` | DAP achieves **the best** trade-offs of the diversity, representativeness and performance|

For the paper writing and presentation, we carefully revised the manuscript to incorporate new experiments, clarify the methodology, highlight key contributions, and address ambiguity and confusion, as suggested by the reviewers. We sincerely thank the reviewers for their valuable feedback.

Finally, to re-emphasize the contributions, DAP achieves state-of-the-art DD performance using only diffusion priors. Additionally, DAP is compatible with other DD works and industrial-level diffusion codebases such as ```diffusers```. It is hoped that this response will assist the ACs in making a comprehensive decision. Thank you for the time and consideration.

Best,

Authors of Paper #11286

---

### Meta-Review · Area_Chair_SWHt · 2026-01-03

**Summary:**

This paper proposes Diffusion As Priors (DAP), a framework for dataset distillation that leverages inherent priors in pre-trained diffusion models. The main concerns are discussion with similar works, design motivation, computational costs, missing implementation details. A rebuttal is provided to address most of these concerns. I am leaning to accept this paper. Author should revise the paper according to discussion.

**Reviewer Concerns:**

Concerns of all reviewers are addressed in the rebuttal.

**Reviewer Scores:**

Reviewer 2soE would not change their score.
Reviewer wTWR may change their score to 6.
Reviewer 1VVS would not change their score.
Reviewer URQY would not change their score.

---

### Decision · Program_Chairs · 2026-01-26

Accept (Poster)